# Pleistocene glacial history of the New Zealand subantarctic islands

Eleanor Rainsley[1]*, Chris S.M. Turney[2,3], Nicholas R. Golledge[4,5], Janet M. Wilmshurst[6,7], Matt S. McGlone[6], Alan G. Hogg[8,9], Bo Li[10,11], Zoë A. Thomas[2,3], Richard Roberts[10,11], Richard T. Jones[12, †], Jonathan G. Palmer[2,3], Verity Flett[13] Gregory de Wet[14], David K. Hutchinson[15], Mathew J. Lipson[2], Pavla Fenwick[16], Ben R. Hines[17], Umberto Binetti[18] and Christopher J. Fogwill[1,2]

1. ICELAB, School of Geography, Geology and the Environment, University of Keele, Staffordshire, ST5 5BG
2. Palaeontology, Geobiology and Earth Archives Research Centre (PANGEA), School of Biological, Earth and Environmental Sciences, University of New South Wales, Sydney, Australia
3. Australian Research Council Centre of Excellence for Australian Biodiversity and Heritage (CABAH), University of New South Wales, NSW 2052, Australia
4. Antarctic Research Centre, Victoria University of Wellington, Wellington 6140, New Zealand
5. GNS Science, Avalon, Lower Hutt 5011, New Zealand
6. Long Term Ecology Laboratory, Landcare Research, Lincoln, New Zealand
7. School of Environment, University of Auckland, Private Bag 92019, Auckland 1142, New Zealand
8. Waikato Radiocarbon Laboratory, University of Waikato, Private Bag 3105, Hamilton, New Zealand
9. Australian Research Council (ARC) Centre of Excellence for Australian Biodiversity and Heritage, University of Waikato, Hamilton 3240, New Zealand
10. Centre for Archaeological Science, School of Earth and Environmental Sciences, University of Wollongong, Australia
11. Australian Research Council (ARC) Centre of Excellence for Australian Biodiversity and Heritage, University of Wollongong, NSW 2522, Australia
12. Department of Geography, University of Exeter, Exeter EX4 4RJ, UK
13. School of the Environment, University of Dundee, Nethergate DD1 4HN, UK.
14. Institute of Arctic and Alpine Research, University of Colorado, Boulder, Boulder, CO, USA
15. Bolin Centre for Climate Research and Department of Geological Sciences, Stockholm University, Stockholm, Sweden
16. Gondwana Tree-Ring Laboratory, PO Box 14, Little River, Canterbury 7546, New Zealand.
17. School of Geography, Environment and Earth Sciences, Victoria University of Wellington, Wellington, New Zealand
18. Centre for Ocean and Atmospheric Studies, School of Environmental Sciences, University of East Anglia, Norwich NR4 7TJ, UK.

† Deceased

*Correspondence to*: Eleanor Rainsley (eleanor.rainsley@gmail.com)

**Abstract.** The New Zealand subantarctic islands of Auckland and Campbell, situated between the Subtropical Front and the Antarctic Convergence in the Pacific sector of the Southern Ocean, provide valuable terrestrial records from a globally-important climatic region. Whilst the islands show clear evidence of past glaciation, the timing and mechanisms behind Pleistocene environmental and climate changes remain uncertain. Here we present a multidisciplinary study of the islands – including marine and terrestrial geomorphological surveys, extensive analyses of sedimentary sequences, a comprehensive dating program, and glacier flowline modelling – to investigate multiple phases of glaciation across the islands. We find evidence that the Auckland Islands hosted a small ice cap at 384,000 ± 26,000 years ago (384±26 ka), most likely during Marine Isotope Stage 10, a period when the Subtropical Front was reportedly north of its present day latitude by several degrees, and consistent with hemispheric-wide glacial expansion. Flowline modelling, constrained by field evidence, suggests a more restricted glacial period prior to the LGM that formed substantial valley glaciers on Campbell and Auckland Islands at around 72-62 ka. Despite previous interpretations that suggest the maximum glacial extent occurred in the form of valley glaciation at the Last Glacial Maximum (LGM; ~21 ka) age, our combined approach suggests minimal LGM glaciation across the New Zealand Subantarctic Islands, and that no glaciers were present during the Antarctic Cold Reversal (ACR; ~15-13 ka). Instead, modelling implies that despite a regional mean annual air temperature depression of ~5°C during the LGM, a combination of high seasonality and low precipitation left the islands incapable of sustaining significant glaciation. We suggest that northwards expansion of winter sea ice during the LGM and subsequent ACR led to precipitation starvation across the mid to high latitudes of the Southern Ocean, resulting in restricted glaciation of the subantarctic islands.

# 1 Introduction

The Southern Ocean plays a critical role in the climate system, connecting the three main ocean basins (i.e. Atlantic, Pacific and Indian Oceans) and modulating regional to global atmospheric temperatures and weather patterns on historic to millennial timeframes (Anderson et al., 2009;Marshall and Speer, 2012;Steig et al., 2009;WAIS, 2015;Turney et al., 2016a;Jones et al., 2016). During the late Pleistocene (110,000 to 11,650 years ago, 110-11.65 ka), marine and terrestrial records across the mid-latitudes demonstrate broadly similar trends to those reported from Antarctic ice core sequences (EPICA, 2006;Kaplan et al., 2008;McGlone et al., 2010;Moreno et al., 2009;Pahnke et al., 2003;Pedro et al., 2015). Following the Last Glacial Maximum (LGM, 21±3 ka), these records show climatic amelioration from ~17 ka through to the onset of the Holocene (Walker et al., 2009) interrupted by a two-thousand year cooling event described as the Antarctic Cold Reversal (ACR), approximately 14.7 to 12.7 ka (Hogg et al., 2016;Pedro et al., 2015).

In the south Pacific, greatest cooling was focussed on ~28-20 ka (the so-called early- or eLGM) (Denton et al., 2010;Fogwill et al., 2015;Hein et al., 2010;Newnham et al., 2007), with sea-surface temperatures (SSTs) apparently between 6-10 ˚C lower than present day (Barrows et al., 2007;Hayward et al., 2008;Pahnke and Sachs, 2006;Panitz et al., 2015), and terrestrial proxies suggesting comparable air temperature reductions of the order of 5-6 ˚C over southern New Zealand (Golledge et al., 2012;Newnham et al., 1989). The prevailing winds and precipitation belts appear to have varied both spatially and temporally with these changes throughout the late Pleistocene (Fogwill et al., 2015;Jaccard et al., 2013). Whilst there is abundant evidence of widespread glaciation across mainland New Zealand during these periods (Porter, 1975;Williams et al., 2015), there are relatively few high-resolution records available from the Southern Ocean (McGlone et al., 2010). The subantarctic islands that span the Southern Ocean have the potential to provide valuable palaeo-climatic and -environmental records. To gain a fuller understanding of climate dynamics over the region it is therefore vital we exploit these currently under-utilised archives fully.

## 1.1 The New Zealand Subantarctic Islands

Located in the Pacific Sector of the Southern Ocean, the New Zealand subantarctic Auckland (50.70°S, 166.10°E) and Campbell (52.54°S, 169.14°E) Islands (Fig. 1) are the eroded remnants of extensive Oligocene to Miocene basaltic volcanism (Quilty, 2007), with rugged topography, reaching up to 569 m above sea level (a.s.l) (Campbell Island) and 664 m a.s.l. (Auckland Island). At present the islands are blanketed with peat deposits (up to 10 m thick) to at least 400-500 m a.s.l. The islands lie in the southwest Pacific in the core of Southern Hemisphere westerly winds and experience the same climate regime with relatively mild mean annual temperatures (Turney et al., 2017). Situated today between the Subtropical Front (approximately 45° S) and the cool Antarctic waters of the Antarctic Convergence (around 55° S), these ocean fronts may have shifted by up to 5° further northwards during the LGM (Nelson et al., 1993) leading to a mean annual air temperature depression over the islands of ~6°C (McGlone, 2002). Campbell and Auckland Islands are therefore highly sensitive to regional climate change during the late Pleistocene, which is potentially archived in their extensive peat deposits and preserved landscape features (Turney et al., 2016b;McGlone, 2002).

The extent to which the subantarctic islands in the southwest Pacific were glaciated during the LGM remains highly uncertain, leaving many questions regarding the late Pleistocene climate in the region unanswered. Previous research carried out on the Auckland Island archipelago reported exposed sediment sections on Enderby Island (Fig. 1) with two distinct layers of glacial till, separated by laminated silt deposited in a pro-glacial lake environment (Fleming et al., 1976). The so-called 'Enderby Formation' was interpreted as having been deposited by the advance of a glacier in the U-shaped valley of Port Ross, and it was therefore assumed that the numerous fjord and cirque features along the east and south coasts of the main Auckland Island also sustained significant glaciation (Fleming et al., 1976;Hodgson et al., 2014a;McGlone, 2002). The early study by Fleming et al., (1976) found pollen and spores in the silt between the two till layers on Enderby Island (Fleming et al., 1976) and suggested this showed there may have been two glacial events with an extended period in between, but this could also represent minor glacier fluctuations of the same event. However, little data have been produced constraining the extent and timing of glaciation on the Auckland Islands; in the absence of a dated chronology these features have been assumed to be evidence of an extensive valley glaciation

occurring during the LGM (Fleming et al., 1976;Hodgson et al., 2014a;McGlone, 2002), although it is also acknowledged that the features are undated and may be of older origins (Hodgson et al., 2014b). Campbell Island, c. 300 km further south and with a comparable elevation, exhibits a similar geomorphology to that of the Auckland Islands, with wide U-shaped valleys and cirque-like features (Oliver et al., 1950). Possible glacial depositional features such as till (overlain by peat with basal dates of ~13,000 yr BP), erratic boulders, and kame terraces have also been reported, posited to be of LGM origin (Hodgson et al., 2014a;McGlone et al., 1997); Campbell (1981), however, suggested some of them may have been formed by fluvial processes. Furthermore, palaeoecological reconstructions suggest that endemic flora and fauna on the islands prevailed throughout the late Pleistocene (Mitchell et al., 2014;Van der Putten et al., 2010;McGlone et al., 2010), conflicting with the paradigm of extensive LGM glaciation (Hodgson et al., 2014a;McGlone, 2002) and raising fundamental questions about our understanding of LGM climatic conditions across this sector of the Southern Ocean.

Here we report the results of a multidisciplinary study of the New Zealand subantarctic Campbell and Auckland Islands that aims to constrain the extent and timing of glacial activity during the late Pleistocene and provide insights into past shifts in climate regimes across the Southern Ocean.

## 2   Methods and study sites

To understand the glacial history of the islands we undertook a multidisciplinary field approach during the Australasian Antarctic Expedition 2013-14 (AAE) and subsequent fieldwork in November 2014, building on previous work led by MM and JW (McGlone et al., 1997;McGlone et al., 2000). This research programme encompassed geomorphological and aerial surveys, an extensive radiocarbon ($^{14}$C) and optical dating program, and the analysis of multiple marine and terrestrial sediment sequences, allowing us to constrain the outputs of targeted glacier flowline modelling. Further technical details relating to the chronological methodology and the modelling can be found in the extended methodology in Appendix 1.

### 2.1 Study sites

Within this study we targeted several areas and sediment formations, located on Figure 1 and briefly described below:

*Lower Lake Speight, Auckland Island:* This cirque, described for the first time by this study (see Sections 2.2 and 3.1 and Figure 2 for more detail), is a relict glacial feature trending east-north-east with significant terminal moraines. Not itself hosting a lake, the cirque is situated above Coleridge Bay in Carnley Harbour, Auckland Island, immediately south of glacial Lake Speight.

*Pillar Rock, Auckland Island:* We report here a new exposed sedimentary sequence on a north-facing cliff near Pillar Rock, Auckland Island (50.518°S, 166.217°E; 30 m a.s.l., Figs. 1 and 3), consisting of a glacial till overlain by a sequence of organic and non-organic units (see Section 3.2 for a full description).

*Carnley Harbour, Auckland Island:* Carnley Harbour is a large sheltered body of water within the Auckland Islands, bordered by Auckland Island to the north and Adams Island to the south. It has three major arms branching off it, including Musgrave Bay.

*Musgrave Harbour, Auckland Island:* This is the land-based cove situated at the head of Musgrave Bay.

*Enderby Island:* This is one of the smaller islands of the Auckland Islands archipelago, situated to the northeast of Auckland Island. The *Enderby Formation* (described in Sections 1.1 and 3.1) is situated on the north coast.

To provide a robust geochronological framework we also undertook an extensive radiocarbon dating programme of new material combined with previously collected sequences (Fig. 1, Tables 1 and 2) to complement previously reported ages (Moar, 1958; McGlone, 2002). The latter sites include the peat sequences from the Auckland Islands at Ranui Cove in Port Ross (Core 8) and two closely adjacent profiles (Deas Head Forest and Deas Head Bog, Core 4) from peats on McCormacks Peninsula, Port Ross. Five other unpublished sites collected by MM and JW in the Port Ross area in 1994 have new basal ages reported here: a bog from Mt Hooker at shrubline (Core 9); blanket peats from the cliffs of North Point Bay, northern Auckland Island (Core 2); a peat sequence adjacent to Pillar Rock, northern McCormacks Peninsula (Core 3); and two cores from a raised bog complex "O", in the interior of McCormacks Peninsula (Cores 6 and 7). A previously published basal $^{14}$C date from Deas Head

(NZA4509; 15,170±140 $^{14}$C yr BP, $\delta^{13}$C -34.1‰) (McGlone, 2002) was obtained from a sandy 'peat', but with ~1% carbon content using this age for the onset of the peat growth is considered highly uncertain (Lowe and Walker, 2000). A new core was therefore taken from Deas Head, and the basal age reported here (Core 4). All samples carbon dated in this study have a carbon content >10%. On Campbell Island, the west-facing Hooker Valley (Core 16) peat and silt cliff section was published in McGlone *et al.* (1997) and the upland Mt Honey Saddle raised bog (Core 21) and Homestead Scarp blanket (Core 18) peat sites were published in McGlone *et al.* (2010). Several other Campbell Island sites have been dated but only the last 2000 years of the sequences published (McGlone et al., 2007). Here we report new basal ages from two of these Campbell Island sites: the lowland raised bog at Homestead Ridge (Core 19) and the upland, west-facing Col Ridge blanket peat (Core 17).

## 2.2 Geomorphological survey

Initial surveying was carried out using topographic maps and satellite imagery to identify possible sites of former glaciation, which were then ground-truthed during fieldwork. Additional sites showing potential glacial features were identified from Fleming et al. (1976), as well as his field diaries (Hince, 2007). Relevant sites were targeted with an Unmanned Aerial Vehicle (UAV) to produce high-resolution digital elevation models (DEMs.) Using a Sensefly 'eBee' mapping drone, imagery was acquired, taking aerial photographs with significant longitudinal and latitudinal overlaps (80% and 85% respectively) to produce a high-resolution (8 cm per pixel) image mosaic and digital surface model. Imagery was fully georeferenced using GPS data and senseFly's eMotion post-processing flight planning software. Digital elevation models (DEMs) of survey sites were compiled using Post Flight Terra 3D 3, and imagery was produced with Global Mapper v16.

To investigate the potential presence of relict glacial geomorphology including moraine features on the inner shelf areas of the New Zealand subantarctic islands, we undertook a ship-based multibeam survey within Perseverance Harbour, Campbell Island (Fig. 1), a valley with an over-deepened U-shaped profile typical of formerly glaciated regions. Multibeam mapping was undertaken with a Kongsberg EM3002 multibeam echo sounder, which was deployed from a side mounted swinging arm. The system

is well-suited for detailed seafloor mapping in water depths from less than 1 metre up to typically 200 metres in cold oceanic conditions. The resultant images were fully georeferenced and simultaneously processed at sub-decimetre accuracy with an integrated differential GPS system.

**2.3 Sedimentary analysis**

Sediment cores were taken from outside the relict glacial limits above Musgrave Harbour, Auckland Island (Core 10, Fig. 1, Figs. S1 and S2, Table 1), and from inside ('LLS', Core 14) and outside ('LLS Terrace', Core 15) the moraine limits of Lower Lake Speight on Auckland Island (Figs. 1 and 2, Table 1). Several other cores were taken from across the Auckland and Cambell Islands (Fig. 1, Table 1) and analysed for basal peat dates. Sediment cores were recovered using a 5-cm wide 'D'-section ('Russian') corer. Organic units at Pillar Rock were sampled with monolith tins. Sediments were described in the field, with samples wrapped and transported back to the laboratory where they were cold-stored (4˚C) prior to analysis.

The LLS and LLS Terrance cores were analysed for organic content by loss-on-ignition (LOI). The LLS Cirque core was subsampled (sample size 0.5 cm) every 2 cm from 64-104 cm and every 1 cm thereafter, focussing on the pre-Holocene section of the core. The LLS Terrace core was subsampled (sample size 0.5 cm) every 5 cm from 0-95 cm, and every 2.5 cm thereafter. These samples were dried overnight in an oven at 105°C and then heated in a muffle furnace at 550°C for 4 hours. The LOI was calculated as the percentage dry weight (Heiri et al., 2001).

Two shallow marine sediment cores were taken from Carnley Harbour (50.82˚S, 166.01˚E, Cores 11 and 12, Fig. 1, Table 1) using a small gravity piston corer with 150cm plastic core barrels deployed over the side of the expedition vessel. The piston corer has an impact-triggered seal that creates a vacuum seal on the core barrel once the corer is retrieved, capturing the water sediment interface and preventing 'wash-out' of fine sediment during retrieval. Over 1 m of sediment was recovered from each deployment of the corer, with the cores consisting mostly of peats overlain by fine silts containing occasional marine mollusk shells in life position.

Samples were taken from the face of the Enderby Formation laminated silt (Fig. 4) at intervals of approximately 10 cm. These were prepared for particle size analysis using a Saturn Digitiser and the results categorised using the Udden-Wentworth Scale (McCave and Syvitski, 1991). The laminated lake sediments from the Enderby Formation were systematically sampled and analysed for pollen, but none was found, in contrast to the earlier study by Fleming et al. (1976)

## 2.4 Chronology

### 2.4.1 Radiocarbon dating

The peat sediments analysed here were given an acid-base-acid (ABA) pretreatment and then combusted and graphitized in the University of Waikato accelerator mass-spectrometry (AMS) laboratory, with $^{14}C/^{12}C$ measurement by the University of California at Irvine (UCI) on a NEC compact (1.5SDH) AMS system. The pretreated samples were converted to $CO_2$ by combustion in sealed pre-baked quartz tubes, containing Cu and Ag wire. The $CO_2$ was then converted to graphite using $H_2$ and a Fe catalyst, and loaded into aluminium target holders for measurement at UCI. To estimate the timing of the onset of peat growth we used the Phase model option in OxCal v.4.2.4 (Bronk Ramsey and Lee, 2013) with General Outlier analysis detection (probability = 0.05) (Ramsey, 2009). See the extended methodology (Appendix 1) for OxCal code. Importantly, the Phase option is a grouping model which assumes no geographic relationship between samples, simply that the ages represent a uniform distribution between a start and end boundary. The $^{14}C$ ages were calibrated against the Southern Hemisphere calibration (SHCal13) dataset (Hogg et al., 2013) and reported here as mean calendar years (yr) or thousands of years (ka) cal BP ± 1σ uncertainty (Tables 1 and 2). Radiocarbon ages are distinguished from calibrated ages by being expressed as $^{14}C$ yr/ka BP.

### 2.4.2 Optical dating

With no organic material present, to provide an age constraint for the Enderby Formation (Figure 1) we collected four samples (Enderby-1, -2, -3 and -4) for optical dating from the sediments overlying the till, hammering opaque tubes (5 cm in diameter) into the cleaned section faces. The tubes were removed and wrapped in lightproof plastic for transport to the Luminescence Dating Laboratory at the University

of Wollongong. Given the potential for the samples to lie beyond the dating range of quartz optically stimulated luminescence, we instead used infrared stimulated luminescence (IRSL), measuring equivalent dose ($D_e$) values for individual potassium-rich feldspar (K-feldspar) grains using a pIRIR (post-infrared IRSL) procedure, made possible by recent developments in the field (Buylaert et al., 2009;Li et al., 2014;Li and Li, 2011;Thiel et al., 2011;Thomsen et al., 2008). Details of sample preparation methods are provided in the extended methodology in Appendix 1, together with $D_e$ and dose rate measurement procedures and supporting data.

## 2.5 Flowline modelling

To investigate the impact of changing climatic conditions on glaciation across the southwest Pacific subantarctic islands during the late Pleistocene, we used a glacier flowline model ensemble approach. Due to the complex topography of the Auckland Islands we focussed our modelling on the west-east aligned Perseverance Harbour (Campbell Island), a U-shaped valley likely to have been an independent ice catchment and ideally suited for flowline modelling (Fig. 1D). An ensemble of experiments was designed that explored the influence of changing a range of parameters on ice extent over the past 125 ka, to identify glacier margin fluctuations within the last glacial cycle.

The glacier model employed is essentially the one-dimensional shallow-ice approximation (SIA) flowline model of Golledge & Levy (2011), with two modifications implemented for this study. Firstly, the positive degree-day (PDD) scheme formerly used to calculate surface mass balance (SMB) has been replaced by a simplified energy-balance scheme, following the insolation-temperature melt method (Eq. 16 of Robinson et al., (2010)). The second modification to the model is the way in which basal sliding velocities are calculated. Previously we used a triangular averaging scheme (Kamb and Echelmeyer, 1986) to smooth calculated driving stress values, which were then used to calculate sliding velocities using a rate factor and sliding exponent (Eq. 8 of Golledge and Levy (2011)). Here we follow instead the approach of Bueler & Brown (2009), and superpose velocity solutions from the SIA with those from the shallow-shelf approximation (SSA).

Input data necessary for our modelling experiments are bed topography (taken from topographic maps) and present-day air temperature and precipitation data (obtained from the National Institute of Water and Atmospheric Research, New Zealand). Time-series data for the study period (125 ka to present) are used for summer and winter insolation values and for air temperature and eustatic sea level perturbations from present. For air temperature anomalies we use two forcing patterns, both scaled to a range of prescribed minima at the LGM. Given the parallel changes across the Southern Ocean region and the Antarctic (e.g. Röthlisberger et al. (2008);Fogwill et al. (2015)), we use the $\delta^{18}O$ record from EPICA Dome C (EDC) (Parrenin et al., 2007) as a proxy for Southern Hemisphere atmospheric temperature changes. In addition, we used a proximal sea-surface temperature (SST) record from core DSDP 594 (Hayward et al., 2008), located to the east of New Zealand's South Island (43.632°S 179.379°E, Fig. 1), rather than that of Pahnke et al. (2003) from core MD97-2120 from the same site. Although the large-scale changes in DSDP-594 closely follow those seen in the EPICA Dome C Antarctic ice core, we acknowledge that a different proxy-based temperature reconstruction, for example from MD97-2120, might produce different glacier behaviour. Starting at 125 ka we initialise our model with present-day topographic and climatological conditions, and simulate glacial advance and retreat according to the calculated SMB and glaciological parameterisation described above. A longer time period was not modelled due to computational cost, as well as the increasing uncertainties associated with all physical boundary conditions.

Since there are few empirical constraints on glacier extent in Perseverance Harbour, our modelling employs full factorial sampling so that results from the mutual combinations of a range of model and climatological parameters can be evaluated with respect to the sparse data, without a priori assumptions about glacier geometry or palaeoclimate. Our parameter-space spans mean annual air temperature (MAAT) scalings of 3, 4, 5, 6, 7, and 8 °C, annual temperature ranges of 5, 8 and 11 °C, and atmospheric transmissivity values of 0.245, 0.25, 0.255, and 0.26 (based on winter values of Robinson *et al.*, (2010) to account for greater year-round cloudiness at Campbell Island than over the Greenland Ice Sheet). We perform each of these parameter combinations both with and without an air temperature / precipitation coupling based on the Clausius-Clapeyron relationship. This relationship parameterizes

the processes that lead to a reduction in moisture-holding capacity as air cools, and equates to a change in precipitation of approximately 7% per degree C air temperature change. Finally, each of the parameter combinations is used in runs forced by each of the EDC and SST timeseries data. The ensemble therefore results in 288 unique simulations. The model is described in detail in Appendix 1.

## 3   Results

### 3.1 Geomorphology

Surveys using satellite imagery, topographic maps and ground-truthing highlight glacial features on both islands, despite the thick peat cover that subdues the glacial geomorphology across much of the islands.

Hydrographic charts hint at the presence of drowned moraine features within the over-deepened valleys of the eastern coasts of both islands, including Norman Inlet on Auckland Island (Fig. S9), and Perseverance Harbour on Campbell Island, which also have a number of promontories shown in the topographic maps that may signify the presence of preserved moraines. Multibeam data show a distinct pair of now-submerged moraines at Shoal and Boyack Points in Perseverance Harbour (Fig. 5), consistent with the over-deepening of the inlet. The terrestrial expression of these features is unfortunately blanketed with thick peat deposits, preventing the identification of glacial sediments, but the extension of these features up the valley sides and their presence mirroring each other on both sides of the harbour strongly suggest they are moraine features. No such features were observed at other promontories such as De la Vire and Davis Points (Fig. 5).  These moraines provide the first evidence of past glaciation within Perseverance Harbour and we suggest that they mark the greatest preserved extent of valley glaciation, at ~6 km from the valley head.

In addition to the above, we report a number of cirques at higher elevations on both Auckland and Campbell islands, with cirque floor elevations of ~115-233 m a.s.l. and maximum of lengths of ~ 1km; the locations of these are shown in Fig. 1. A UAV survey of one of the most distinct cirques, Lower 'Lake' Speight on Auckland Island (LLS, immediately south of glacial Lake Speight but not itself

hosting a lake) captures the geomorphological feature in detail; the cirque is over-deepened, with a vertical back wall and distinct terminal moraines, with no evidence of nested moraines within their limits (Fig. 2). Unfortunately, extensive vegetation or thick peat deposits extending to the highest altitude of the islands bury any potential erratic boulders or glaciated bedrock at our surveyed sites, precluding direct dating of these features, or in-depth geomorphological mapping of the cirque sites. No other terrestrial glacial geomorphology outside the limits of these cirques was observed on either the Auckland or Campbell islands.

## 3.2 Sediment sequences

The glacial till described by Fleming *et al*. (1976) on Enderby Island appears to also be represented in similar extensive drift deposits up to 6 m thick, which we identified at Pillar Rock on the north coast of Auckland Island and along the shoreline at Emergency Bay within Carnley Harbour (Fig. 1B and C, Fig. S7A-C and S8). For ease of reference, we refer to this till as 'Enderby Till', wherever it is found on the island. The Enderby Formation outcropping on Enderby Island is the most complete exposure of glacial sediments on the islands, and in common with all other outcrops of the sequence is highly indurated. As described by Fleming et al. (1976), the formation consists of a lower diamict (Enderby Till Fig. S8A) overlain conformably by massive sands and silts which grade upwards into laminated silts with dewatering structures. Above the laminated silts the sequence becomes coarser, with an increasing percentage of sand evident in laminations which grade into sandy gravels and finally laminated gravels that grade into an overlying upper diamict (Enderby Till). The laminated silts and sands are not found elsewhere on the islands, but the Enderby Till diamict is mirrored at Pillar Rock and Emergency Bay, (Fig. S7 and S8), suggesting that the exposures at Pillar Rock and Emergency Bay record the same glacial event.

The pIRIR ages of the laminated sediment in the Enderby Formation are listed in Table 3 and presented in Figures 4 and 9. All four ages have large relative uncertainties (10–35% at $1\sigma$), mainly because of the small number of grains accepted for $D_e$ estimation, but they are consistent at $2\sigma$ with a common value. The pIRIR ages for the four samples are 400±64, 343±43, 284±101, 413±39 ka for Enderby-1, -2, -3

and -4, respectively. Enderby-1 and especially -3 have large age uncertainties due mainly to the small number of grains (< 10 grains) suitable for $D_e$ determination. The samples are statistically indistinguishable from each other and are effectively from a single unit/event. We therefore compute the weighted-mean age (excluding 'Enderby-3', which we consider to be an outlier) as 384±26 ka. Inclusion of Enderby-3 gives a weighted mean age of 378±26 ka; either way, the difference is well within the errors and does not affect our interpretation. This spans the time interval from about 330 to 440 ka at the 95% confidence interval, corresponding to Marine Isotope Stages 9 to 12, considerably older than the previously assumed LGM age.

The Pillar Rock section provides important further constraints to Pleistocene glaciation on the Auckland Islands. The exposed section (Fig. 3) consists of a laterally continuous (~10 m), horizontally bedded section exposed by extensive cliff erosion along the north coast of Auckland Island at ~30 m.a.s.l. (Fig. 1C). In places the section is over four m thick, and rests on several metres of deeply indurated diamicton, that we correlate with the upper till exposed on Enderby Island. The 'Enderby Till' at Pillar Rock is overlain non-conformably by stratified sands and gravels, which are iron banded. This unit is overlain by a ~2.4 m thick conformable sedimentary succession, that grades upwards from massive silts and sands into laminated silts and clays and into an organic-rich silt band (Fig. S7D). We have dated this organic-rich silt band, with the lower boundary, at a depth of 250 cm in the sequence of 42,230±1200 [14]C yr BP (45,900±1250 cal yr BP), and at the upper boundary, at a depth of 237 cm, to 36,860±600 [14]C yr BP (41,350±500 cal yr BP). Above this organic silt the succession continues, with a conformable, non-erosive boundary into sandy silts, which become increasingly clast rich and less dense. This conformable sequence is topped by a 10 cm thick clast-supported gravel, topped by a 2 cm layer of flat lying gravel / pebbles. This succession is overlain unconformably by a 27 cm thick sequence of humified dark brown peat, dated between 10,090±30 [14]C yr BP (11,550±120 cal yr BP) and 8110±30 [14]C yr BP (8960±80 cal yr BP) at the lower and upper boundaries respectively. Above the peat, there is a series of horizontally bedded silty sandy loams and low organic soil horizons, with pebble horizons, overlain by ~20 cm of sandy loam at the surface.

The two marine sediment cores from Carnley Harbour (Cores 11 and 12, Fig. S2) gave basal peat ages of 12,620±50 cal yr BP and 11,690±130 cal yr BP. Calibrating the basal radiocarbon ages of peat from across Campbell and Auckland Islands for all sequences with [14]C ages exceeding 9 [14]C ka BP – including the Carnley Harbour cores – provides an age constraint for the onset of peat growth across the islands (Table 1). The OxCal Phase age calibration of the basal peat samples suggests the onset of peat growth commenced at 17,570-15,580 cal yr BP at 2σ (mean of 16,610±390 cal yr BP or 16,820-16270 cal yr BP at 1σ).

Sediment sequences from key cores both within and directly outside the limits of key cirques (LLS Cirque, LLS Terrace, and Musgrave Harbour, Fig. 1) were dated throughout their profiles and analysed for organic content using LOI. Organic lake mud from the base of the Musgrave Harbour core (Core 10, Fig. 1, Table 1) at 175 cm is dated to 10,340±60 cal yr BP. LOI and radiocarbon dating from a core from within the LLS cirque (Site 14, Figs. 1 and 6, Table 2) suggests onset of peat growth in the cirque commenced at 11,700±140 cal yr BP; however, an age inversion in the silty material further up the core suggests a relocation of young carbon along the contact zone between the silt and organic layers, and this constraint is therefore questionable. The LLS Terrace core, however, gives a coherent age-depth profile (Site 15, Figs. 1 and 6, Tables 1 and 2), and shows that organic matter content in the outwash zone of the LLS Cirque increased markedly between 12,310±50 and 11,950±40 [14]C yr BP (14,200±120 and 13,710±70 cal yr BP) with no evidence of a reversal in trend. No glacial material was found at the base of the cores.

### 3.3 Flowline modelling

The 288 model runs span a large range of climatic conditions under which Perseverance Harbour (Campbell Island) could have been glaciated during the late Pleistocene. Figure 7A illustrates the glacier-length predictions of all 288 model runs, coloured according to the MAAT depression used to scale the EDC or SST time series forcing. Simulations that extended beyond the full length of Perseverance Harbour were terminated, consistent with our multibeam survey, which suggested the limits of Late Pleistocene glaciation were well within the constraints of the Harbour; this does not

preclude the possibility of extensive glaciation across the island during an earlier period. To constrain the configurations that represent realistic scenarios, the comprehensively radiocarbon-dated peat sequences from Homestead Bog, Campbell Island (Fig. 1D, Table 1) (McGlone et al., 2010;Turney et al., 2006) were used as a constraint. At 40 m a.s.l. on the northern shore of Perseverance Harbour, 3.8 km from the head of the flowline (Fig. 1D, Table 1), the presence of peat in the valley indicates glacier absence from ~15 ka through to present. We allowed a 5% tolerance to account for a glacier snout to be slightly more extended in the centre of the valley than on the flanks where the [14]C dates were obtained, and to account for any spatial or temporal uncertainties not explicitly accounted for elsewhere. This constraint allowed us to select those simulations that model glacier length to less than 4 km at 15 ka.

The selection described above results in a suite of 102 simulations (Fig. 7B). Of these, the large majority (69%, Fig. 7B inset) – including those which model unrealistic glacial lengths during the Last Interglacial – occur with MAAT depressions that are either smaller than 5 degrees or greater than 7 degrees. These values are inconsistent with the LGM cooling constrained to ~6°C on Auckland and Campbell (McGlone, 2002). By excluding runs with 3, 4 and 8°C MAAT depressions, and those that take no account of the temperature control on precipitation, we further reduced our sample subset to 25 simulations (Fig. 8 and Fig. S10). Each of these runs produced a different pattern of glacier advance and retreat. By taking an ensemble modelling approach and comparing these patterns with the geomorphological and chronological data available from the last 40 ka, it is possible to refine our interpretation of the most likely climatic conditions that prevailed across the New Zealand subantarctic islands during the last glacial cycle.

## 4   Discussion

Our results show unequivocal evidence of past glaciation, but challenge previous suggestions as to the timing and extent of ice during the LGM (Fleming et al., 1976;Hodgson et al., 2014a;McGlone, 2002). Here we propose the sequence of late Pleistocene events that best explains our field and modelling datasets from the New Zealand subantarctic Campbell and Auckland Islands.

## 4.1 The 'middle' Pleistocene

We interpret the Enderby Formation as a glacial succession that records an oscillating ice margin, with two extensive deposits of subglacial diamicton separated by deposition of ice-proximal glacifluvial sedimentation that was overrun by the later ice advance evidenced by the upper diamicton (Fig. 4, Figs. S7B and S8A). The conformable nature of the sequence suggests that there was not a significant hiatus between the two glacial advances recorded by the diamictons; rather the glaciofluvial sediments sandwiched between the two diamictons were formed during a brief period of recession and subsequent advance during a sustained glaciation. The location of this glacial till on the north shore of Enderby Island has led to previous studies (Fleming et al., 1976;Hodgson et al., 2014a;McGlone, 2002) suggesting this was deposited by a valley glacier emanating from Port Ross, Auckland Island. However, this study has identified similar till at multiple sites across the Auckland Islands; at Pillar Rock these deposits (stratigraphically below those described in Fig. 3) are found exposed in high coastal cliffs, with those at both Pillar Rock and Emergency Bay (Fig. 1) located outside valley settings. This suggests a period of extensive glacial cover of the Auckland Islands (henceforth referred to as the 'Enderby Glaciation'), perhaps as a small ice cap, far beyond the scope of the valley glaciers previously suggested to have been the greatest extent of glaciation on the islands.

The highly indurated nature of the Enderby Formation suggests considerable antiquity, but in the absence of direct chronological control the exposure has been interpreted to be LGM in age. This is overturned by the IRSL dating of the Enderby Formation glaciofluvial sediments to 384±26 ka (Table 3). This places the Enderby Glaciation firmly in the middle of the Pleistocene, most likely during MIS-10 (~374 ka), or potentially MIS-12 (~424 ka), both significant glacial stadials. It is possible that the tills were deposited in two separate glacial events (i.e. the lower in MIS-12 and the upper in MIS-10), but on balance we consider that the conformable appearance of the sediment boundaries and the lack of pollen and spores in the laminated silt supports the likelihood that the entire Enderby Formation was formed in one (extensive) event during MIS-10.

The timing of this glaciation is consistent with other locations in the Southern Hemisphere. Notably, recent work on the Boco Plain in Tasmania (Augustinus et al., 2017) has identified a succession of glacial advances, the most recent of which (prior to the LGM) was 378±22 ka during Marine Isotope Stage 10 (the Boco Glaciation). Intriguingly, the recognition of two lateral moraines, one overlying the other, suggests an oscillating ice margin similar to that observed on Enderby Island, and may represent a common regional climate-driver. Similarly, [10]Be-dated glacial moraines in the Lago Buenos Aires (Argentina) region also indicates maximum glaciation prior to the LGM between ca. 450 and 340 ka (Augustinus et al., 2017;Kaplan et al., 2005).

The hemispheric-wide nature of this major glaciation appears to be associated with extreme southern glacial stadials between 420 and 340 ka (Marine Isotope Stages 10 and 12) (Bard and Rickaby, 2009a;Li et al., 2010). Off southern Africa, a SST reconstruction suggests that the Subtropical Front (and the associated westerly winds) migrated north by some 7° of latitude during both glacial periods, limiting the leakage of the Agulhas Current (warm, saline water) into the south Atlantic Ocean, weakening the Atlantic Meridional Overturning Circulation and amplifying the severity of glaciations in the Southern Hemisphere and possibly globally (Bard and Rickaby, 2009b), the opposite to that suggested for periods of super-interglacial warmth (Turney and Jones, 2010). SSTs during MIS-10 and -12 were 6°C cooler than the Holocene, compared to a depression of just 4°C at the LGM/MIS-2. Given the numerous glacial advances between ~370 and 390 ka across the mid to high-latitudes of the Southern Hemisphere, and our weighted-mean age of 384±26 ka for the laminated sediments, we favour attributing the Enderby Formation to MIS-10.  Regardless of the glacial stage attribution, our results from the New Zealand subantarctics imply that extensive glacial deposits on other subantarctic islands cannot be assumed to be LGM in age (Hodgson et al., 2014b;Balco, 2007) and may similarly be of middle Pleistocene origin.

## 4.2 The Late Pleistocene at Pillar Rock, Auckland Island

The interpretation of the rare exposed sedimentary sequence at Pillar Rock and the chronological control it affords is crucial to the glacial history of the islands, given that at present, the interpretation

rests on the previously held assumption that the lower indurated glacial diamicton, the 'Enderby Till', is of LGM age. The stratigraphy and chronology of this sequence suggest this cannot be the case. The presence of a conformable succession of unindurated soft sediments at Pillar Rock overlying the 'Enderby Till' and clearly predating the LGM is important. The depositional environment of the sequence is a relatively flat coastal plain, currently peat bog; we therefore interpret this unit as capturing the gradual infill of a depression with minimal slope processes at work, most likely during the last glacial cycle. Lower unstratified sands, prograding up into laminated silts and sands, are overtopped by a ~13 cm thick organic silt band, which has calibrated ages of ~46 cal ka BP and ~41 cal ka BP, eliminating the possibility that the underlying Enderby Till is coeval with the LGM. We suggest that this organic unit was formed under anoxic conditions, as the basin or depression filled and ponded water and organic material over several millennia, consistent with enhanced westerly airflow over Auckland Islands and the wider Southern Ocean (Jaccard et al., 2013), and and may correspond to Antarctic Isotopic Maximum (AIM) 12, which represents significant warming in the EPICA Dome C Core in East Antarctica (EPICA, 2006) (Fig. 9).

The Pillar Rock sequence preserves a record of ongoing sedimentation, most likely during the LGM, with the upper layer of the succession being indicative of a winnowed lag deposit, formed under cold, periglacial conditions. This interpretation is supported by the presence of an overlying thick humified peat deposit, lying unconformably above this succession, formed between ~11.6 cal ka BP and ~9 cal ka BP, during the early Holocene climatic amelioration. The lack of peat above this unit suggests the area had become free draining soon after ~9 cal ka BP, most likely due to erosion of the cliffs surrounding the site of the basin or depression and led to a change in the depositional environment, alternating between periods of soil formation and low organic sedimentation at this exposed location. This could be caused by increasing strength in the westerly winds throughout this period, which may have increased cliff erosion and led to the transfer of sands and silts inland (McGlone, 2002).

**4.3 Evidence for limited Late Pleistocene glaciation?**

The geomorphological evidence reported here suggests that at least one glaciation took place subsequent to the Enderby Glaciation. High-altitude cirques, with pronounced terminal moraines and no evidence of nested inner moraines, show a period of limited but sustained glaciation; organic lake sediments from outside the limits of the Musgrave Harbour cirque dated to ~10 ka cal BP suggest that there was no glacial sediment input from this time. Most significant is the dated sediment sequence from the LLS Terrace core (Core 15, Fig. 1, Table 2), directly outside the limit – and within the sediment outwash zone – of the LLS Cirque. The sharp increase in organic matter content observed in the Terrace core (Fig. 6) continues uninterrupted through the Antarctic Cold Reversal (ACR; 14,500-12,500 cal yr BP (EPICA, 2006)), which was a period of extensive glacier growth in mainland New Zealand and Patagonia (Fogwill and Kubik, 2005;Pedro et al., 2015;Putnam et al., 2010). That the LLS Terrace core shows no inorganic sediment input throughout this period strongly indicates that there was no substantial ACR glaciation in the cirque above this site. This interpretation corresponds with temperature reconstructions from Campbell Island (McGlone et al., 2010) (Fig. 9), which show minimal temperature depression during the ACR. The mean age for the onset of basal peat growth across Auckland and Campbell islands shows that significant climate amelioration had commenced by ~16.6 ka cal BP (Fig. 9), signifying the end of the periglacial conditions across lower latitudes of the islands, and constraining the widespread onset of peat growth, which continued uninhibited by subsequent post-glacial cool periods, including the ACR. Furthermore, peat dated to ~12.6 ka cal BP from two marine cores taken from Carnley Harbour (Cores 11 and 12, Table 1) show that sea level was significantly lower and that this low-lying area was ice-free at the time. It is possible that the later onset of peat growth and imperfect age models within the cirques were caused by the presence of perennial snow patches throughout the post-glacial and early Holocene period (Watson, 1966); however, it is clear that the expression of the ACR was limited over the subantarctic islands in the southwest Pacific, and did not result in glaciation.

We suggest that the lack of evidence for ACR glaciation strongly implies that the pronounced cirques observed across the Auckland and Campbell Islands (Fig. 1) originated during an earlier glacial; we suggest this to be either the LGM or Southern Hemisphere eLGM. There are two potential scenarios for

this cirque glaciation: one, a limited valley glaciation of the islands during the eLGM/LGM, followed by retreat to the cirques before final deglaciation; or two, the cirques represent the maximum extent of eLGM/LGM glaciation. Given the lack of glacial geomorphology observed outside the limits of the cirques, we suggest the latter scenario is more likely, but more work is needed to confirm this interpretation. It must be noted that as the cirques (and associated features i.e. moraines) are not directly dated, it is possible that cirque formation predates the eLGM/LGM. Regardless of which scenario is correct, however, it is clear that glaciation of Campbell and Auckland Islands during the LGM was far more restricted that previously supposed (Hodgson et al., 2014a;McGlone, 2002;Fleming et al., 1976).

**4.4 Modelling Late Pleistocene glaciation**

The Perseverance Harbour moraines revealed on Campbell Island by the multibeam survey indicate the presence of a glacier terminating 6 km from the head of the valley and close to sea level; however, although more extensive than the LGM Auckland and Campbell cirques, this valley glaciation is on a much smaller scale than the Enderby Glaciation. The high level of preservation of these moraines strongly implies that another glacial event in the New Zealand subantarctic islands occurred prior to the LGM but after the Enderby Glaciation. To test this, and the hypothesis that the LGM glaciation was limited, we ran a series of flowline model experiments. 25 model runs of Late Pleistocene glaciation in Perseverance Harbour fit the constraints of the field data (Fig. 8 and Fig. S10), most importantly, the constraint from Homestead Bog, situated at 3.8 km from the head of the model flow line, and shown to be deglaciated by at least ~15ka. The vast majority of these 25, however, reach a maximum extent of just 3-4 km, which is incompatible with the length of glacier needed to form the Perseverance Harbour moraines we report at ~6 km from the valley head in Campbell Harbour. Importantly, though, one model run does capture a significant glacier advance to >5 km at 67.7 ka (Fig. 8A), followed by retreat to 2.3 km and a subsequent limited advance to 3.2 km during the LGM, before rapid deglaciation at 15 ka prior to the ACR (Fig. 8A).

We suggest, therefore, that the Perseverance Harbour moraines (and similar valley glaciers on the east coast of Auckland Island) were formed around 68 ka, consistent with the New Zealand glacial maximum of the so-called Otira Glaciation at 62-72 ka during MIS-4 (Williams et al., 2015). In the absence of direct dating we cannot rule out an earlier formation date for these moraines (i.e. during the interglacials of MIS 6 or 8), but the high level of preservation in a submarine setting, together with the agreement with the model simulations, support a more recent event. The model further suggests that the islands were incapable of hosting a more extensive glaciation than this at any point over the past 125 ka, consistent with our interpretation that it took a step-change in the regional climate system to support the extensive MIS-10 Enderby Glaciation. This may have been further exacerbated by the loss of catchment to the west through marine erosion and glacial attrition or tectonic subsidence (Summerhayes, 1967), which may have diminished the available accumulation area for more 'recent' glacials, further reducing the capability of the New Zealand subantarctic islands to sustain glaciation on the scale seen during MIS-10. Quantifying both this potential loss, and the changes to global and regional sea level that may also have affected the catchment of the New Zealand subantarctic islands would be an important next step in determining the extent and nature of MIS-10 glaciation in the region.

## 4.5 Late Pleistocene climatic implications

Given the close correspondence of the field data, we therefore propose that the climatic conditions of the model run shown in Fig. 8A provide a reasonable first-order representation of change over the New Zealand subantarctic islands during the late Pleistocene. For the LGM, our study implies a MAAT depression of 5°C, a similar level of cloudiness as today, an annual temperature range of 11°C, and a substantial precipitation reduction of ~30%. Crucially, seasonality appears to have played an important role in glaciation on these islands; our simulation with identical parameters to the above but with less severe winter cooling (Fig. 8B) fails to reach the glacier length necessary to form the Perseverance Harbour moraines.

The same MAAT depression of 5°C obtained by the colder summers and milder winters of the modelled scenario in Fig. 8B would have placed considerable pressure on the viability of tall shrubs, which would

never have reached the required number of degree growing days to sustain growth. Crucially, the high seasonality implied by the 11°C temperature range invoked in the modelled scenario in Fig. 8A – biased towards the winter – would have allowed for the continued survival of plant life on the islands suggested by the high number of endemic species (McGlone et al., 2010;Mitchell et al., 2014;Van der Putten et al., 2010). The milder summers in this scenario means that the lowland shrubs and herbs found across the islands would have been able to survive through the colder winters. Some woody species (e.g. *Dracophyllum* spp. and *Myrsine divaricata*) are likely to have survived through the LGM on both Auckland and Campbell Islands as they both appear very early in the postglacial period and can be shown to have been present pre-LGM on Auckland Island (Fleming et al., 1976;McGlone, 2002). The current absolute limit to shrubland containing these species on Campbell Island is c. 300 m (McGlone et al., 1997), which equates to a warmest month temperature of ~7°C. If allowance is made for a fall in mean sea level of around 120 m, the maximum depression in MAAT at the LGM that would permit these shrubs and small trees to survive is ~3°C. As pointed out by McGlone (2002) there is a mismatch here with previous estimates of LGM cooling of ~6°C as such a temperature depression, if it affected summer and winter equally, would exceed the tolerance of these woody species. However, a MAAT depression of 5°C, combined with an increased warmest month/coolest month temperature range of 11°C (versus the current 4.7°C), would not only ensure the survival of these woody species but also other species including the functionally flightless Campbell Island and Auckland Island teal (*Anas nesiotis* and *A. aucklandicus*) which, on the basis of molecular dating, have occupied the islands over several glacial-interglacial cycles (Mitchell et al., 2014).

We therefore propose that reduced winter precipitation and high seasonality shown by our flowline modelling combined to limit the LGM glaciation of the New Zealand subantarctic islands despite the large MAAT depression. This is consistent with work showing an expansion of winter sea ice in the Southern Ocean, reducing the moisture available to the atmosphere in this region (Fogwill et al., 2015;Otto-Bliesner et al., 2006). Whilst palaeo records of Antarctic sea-ice extent are incomplete, our flowline modelling is consistent with most proxy and model data, which suggest a significant northwards expansion of sea ice to within the region of the New Zealand subantarctic islands both

during the LGM (Roche et al., 2012) and the ACR (Pedro et al., 2015) with a strong seasonal cycle (Gersonde et al., 2005). These studies are further supported by genetic analysis of kelp across the Southern Ocean that shows that kelp recolonized the subantarctic islands, including Campbell and Auckland Islands, relatively recently, a finding that has been interpreted to suggest the presence of sea ice across the region at the LGM (Fraser et al., 2009). Complementary studies on atmospheric circulation during the LGM suggest an equatorward shift with a specific focus on mid-latitude westerly airflow across this region (Hesse, 1994;Kohfeld et al., 2013;Jaccard et al., 2013); this could have reduced sea ice break-up and storm frequency over the subantarctic islands, further reducing precipitation. Furthermore, reconstruction of palaeo SSTs in the southwest Pacific Ocean suggest that the Subtropical Front underwent a considerable southwards shift during the LGM to ~50°S in the New Zealand region (Bostock et al., 2015), almost reaching the latitudes of the Auckland and Campbell Islands. This is in opposition to the northwards shift during MIS-10 and -12, and further explains the vast disparity in glacial extent during these periods. The suggestion of a restricted LGM of subantarctic islands is not exclusive to the New Zealand region. Southwest Pacific Macquarie Island (54.62°S) and south Atlantic South Georgia (54.25°S) may have experienced similarly restricted glaciation during the late Pleistocene (Bentley et al., 2007;Hodgson et al., 2014a), and Stewart Island (47°S) to the south of the New Zealand mainland also had a very limited LGM extent compared to mainland New Zealand (Brook, 2009), whilst the Kerguelen Islands (49°S) were undergoing active deglaciation from throughout the LGM (Jomelli et al., 2017;Jomelli et al., 2018); this suggests this mechanism of high seasonality and low winter precipitation driven by expanded sea ice may have caused reduced LGM glaciation of the subantarctic islands across the Southern Ocean.

## 5   Conclusions

Palaeo-archives and modelling from the New Zealand subantarctic islands provide important new insights into the glacial history and changing climate of the Pacific sector of the Southern Ocean. We find evidence for multiple distinct phases of glaciation on both Auckland and Campbell Islands, and hypothesize the following sequence of events: 1) the 'Enderby Glaciation' at ~380 ka consisting of extensive glaciation covering large portions of the islands (potentially as small ice caps), most likely

occurring during MIS 10; 2) a more restricted glacial period prior to the LGM that formed substantial valley glaciers on Campbell and Auckland Islands at around 72-62 ka; and 3) a limited eLGM/LGM glacial advance, comprising of either valley glaciation with glaciers no longer than a few kilometers prior to retreat to high-altitude cirques, or of severely restricted glaciation limited to these cirques. We find no evidence for glaciation on the islands after the LGM.

The disparity between limited late Pleistocene glaciation of the New Zealand subantarctic islands and the widespread expansion of glaciers across southern Pacific landmasses (such as Patagonia, Tasmania and New Zealand) raises important questions about our understanding of changing climatic conditions across the Southern Ocean. The Auckland and Campbell Islands were apparently incapable of sustaining significant glaciation during the LGM or ACR despite the pronounced and sustained atmospheric and oceanic temperature depressions, suggesting other factors had a greater influence than temperature changes alone. Our results are consistent with precipitation starvation caused by the northwards expansion of winter sea ice in the Southern Ocean, which overwhelmed regional atmospheric and oceanic temperature depressions and resulted in limited glaciation on the subantarctic islands, providing important new insights into the significance of sea-ice extent and precipitation regime shifts across the Southern Ocean. In marked contrast, during the middle of the Pleistocene, the minimum in eccentricity (100 kyr and 400 kyr cycles) appears to have driven a northward migration of the Subtropical Front, in opposition to the LGM southwards shift. The resulting substantial reduction in Agulhas Current leakage into the Atlantic Ocean depressed southern SSTs over an extended period, sufficient to sustain significant glaciation on the low-altitude New Zealand subantarctic islands in MIS-10. The remarkably disparate responses of glaciation on the Auckland and Campbell Islands to differing regional conditions during the Pleistocene highlights the sensitivity of the Southern Ocean region, and suggests that step-changes in atmospheric and oceanic regimes can have extreme climatic and environmental impacts.

**Appendix 1 – Extended Methodology**

## 1. Optical dating

Each sample was treated using routine procedures to extract sand-sized grains of K-feldspar (Aitken, 1998), treated sequentially with HCl acid and $H_2O_2$ to remove carbonates and organic matter, and then dried. Grains in the size range of 90–125 $\mu$m (Enderby-OSL2, -OSL3 and -OSL4) or 180–212 $\mu$m (Enderby-OSL1) were isolated by dry sieving, and the K-feldspar grains separated from heavier minerals using a sodium polytungstate solution with a density of 2.58 g/cm$^3$. K-feldspar grains were etched using 10% HF acid for 40 min to clean their surfaces and reduce the thickness of the alpha-irradiated outer layer of each grain. The IRSL measurements were made on an automated Risø TL-DA-20 reader using both single-aliquot and single-grain techniques. Aliquots consisting of several hundred grains were prepared by mounting the grains as a monolayer (~5 mm in diameter) on stainless steel discs, using 'Silkospray' silicone oil as the adhesive, and were stimulated using infrared diodes (870 $\Delta$ 40 nm, 135 mW/cm$^2$). For the single-grain measurements, we used discs drilled with 100 holes, each 300 $\mu$m in diameter and depth, and stimulated the grains individually using a focussed infrared laser (830 $\Delta$ 10 nm, 400 W/cm$^2$) (Bøtter-Jensen et al., 2003). Radiation doses were given using a $^{90}$Sr/$^{90}$Y beta source mounted on the reader and calibrated for both multi-grain aliquots and individual grain positions. The IRSL signals were detected using a photomultiplier tube, after passing through a filter pack containing Schott BG-39 and Corning 7-59 filters to transmit wavelengths of 320–480 nm. We used single aliquots to conduct laboratory tests of anomalous fading, but estimated the $D_e$ values from measurements of individual grains of K-feldspar (Duller, 2008;Jacobs and Roberts, 2007), enabling the identification and elimination of any grains with aberrant luminescence characteristics (David et al., 2007;Feathers, 2003;Feathers et al., 2006;Jacobs et al., 2006;Jacobs et al., 2011;Jacobs et al., 2008;Roberts et al., 1998;Roberts et al., 1999) prior to age determination. We did not apply a residual-dose correction, because our samples have high natural doses (>400 Gy).

In this study, we adopted a two-step single-grain pIRIR regenerative-dose procedure (Blegen et al., 2015;Guo et al., 2016) to determine the $D_e$ values for individual sand-sized grains of K-feldspar. In this procedure (see Table S1 for details), an initial infrared stimulation is made at 200 °C for 200 s using infrared diodes, so that all 100 grains on each single-grain disc are stimulated simultaneously. Li and Li

(2012) showed that, compared to an initial infrared stimulation at 50 °C, the fading component is removed more effectively at a temperature at 200 °C. The pIRIR signals used for $D_e$ determination were then measured for individual grains at 275 °C, with each grain stimulated by an infrared laser for 1.5 s; this duration proved sufficient to reduce the pIRIR intensity to a low and stable level (Fig. S3). We chose a stimulation temperature of 275 °C, rather than 290 °C (Thiel et al., 2011), to avoid any detrimental effects from significant thermal erosion during the single-grain measurements. The latter can occur at stimulation temperatures close to the preheat temperature (320 °C), because laser stimulation of the final grain on each 100-grain disc begins more than 2.5 min after the first grain on the disc. Figure S3 shows a typical pIRIR decay curve and associated dose response curve for a single grain of K-feldspar from Enderby-OSL1.

We conducted a dose recovery test (Galbraith et al., 1999; Roberts et al., 1999) on Enderby-OSL1 to validate the suitability of the pIRIR experimental conditions used to measure these samples (Table S1). A total of 600 grains were bleached for ~6 hr using a Dr Hönle solar simulator (model: UVACUBE 400) and then given a dose of ~300 Gy, before being measured using the procedure of Table S1. To select reliable single grains for $D_e$ determination and reject unsuitable grains, we applied quality-assurance criteria similar to those proposed for single grains of quartz (Jacobs et al., 2006) and K-feldspar (Blegen et al., 2015). Grains were rejected if they exhibited one or more of the following five properties: 1. Weak test-dose signal; 2. High level of recuperation; 3. Poor recycling ratio; 4. Poorly fitted dose response curve (DRC); or 5. Natural pIRIR signal equal to or greater than the saturation limit of the DRC. We found that most of the grains (583 grains) were rejected by criterion 1 because of weak $T_n$ signals. Among the remaining 17 grains, two had $L_n/T_n$ ratios greater than the saturation limit of the DRC, two had poorly fitted DRCs, and one grain suffered from high recuperation. As a result, only 12 grains were accepted for dose estimation. The measured-to-given dose ratios for these grains are shown in Fig. S4a. The weighted mean of these 12 dose recovery ratios is 1.06 ± 0.07, which is consistent with unity and suggests that the pIRIR procedure in Table S1 yields reliable $D_e$ values for the Enderby samples. We also tested the Enderby samples for anomalous fading of the pIRIR signal. Twelve aliquots of Enderby-OSL1 were measured using a single-aliquot IRSL procedure similar to that

described by Auclair et al. (2003), but based on fading measurements of the pIRIR signal. As our samples emitted relatively dim signals, we used the procedure of Thiel et al. (2011), in which the initial infrared stimulation is made at 50 °C and the pIRIR signal is subsequently stimulated at 290 °C. Doses of ~200 Gy were administered using the laboratory beta source, and the irradiated aliquots were then preheated and stored for periods of up to 1 week at room temperature (~20 °C). Figure S4b shows the decay in the sensitivity-corrected pIRIR signal as a function of storage time, normalised to the time of prompt measurement. The corresponding fading rate (*g*-value: (Aitken, 1985;Huntley and Lamothe, 2001) is 0.6 ± 0.7% per decade, which is consistent with zero fading at 1σ. Li and Li (2012) demonstrated that a higher initial infrared stimulation temperature (e.g., 200 °C) could remove the fading component more effectively than stimulating at 50 °C, so the pIRIR signal measured after an initial infrared bleach at 200 °C (Table S1) should be stable. Hence, no fading correction has been made to the measured $D_e$ values for the Enderby samples.

We consider the $D_e$ values obtained for all samples as effectively single-grain estimates. Grains of 90–125 μm diameter were used for Enderby-OSL2, -OSL3 and -OSL4, so each hole on the disc may contain ~8 grains for these samples. For Enderby-OSL1, each hole was occupied by a single grain of 180–212 μm in diameter, and more than 95% of the total pIRIR signal was contributed by fewer than 5% of the measured grains (Fig. S3b). For Enderby-OSL2, -OSL3 and -OSL4, therefore, we deduce that the measured pIRIR signal arises from only one or two grains in each of the holes.

Given the paucity of K-feldspar grains available for each of the Enderby samples, only 600, 600, 300 and 500 grains were measured for Enderby-OSL1 to -4, respectively. After applying the same five rejection criteria as were used in the dose recovery test, we found that ~20% of the accepted grains had natural signals equal to or greater than the saturation limit of their DRCs; finite $D_e$ values could not be estimated for these grains. As a result, we expect that the $D_e$ estimates for the remaining accepted grains will likely underestimate the true $D_e$ value for each sample, because the rejection of the saturated and 'oversaturated' grains will lead to a truncated $D_e$ distribution (Duller, 2012;Guo et al., 2017;Li et al., 2016;Thomsen et al., 2016). To avoid this problem, we applied the new method proposed by Li et

al.(2017), in which a standardised growth curve (SGC) is established and the weighted-mean $L_n/T_n$ ratio for all accepted grains is projected on to the corresponding SGC to determine the sample $D_e$. In this method, no grains are rejected due to saturation issues, so a full (untruncated) distribution of $L_n/T_n$ ratios is obtained. Based on their experimental and simulation results, Li et al. (Li et al., 2017) showed that this method can produce reliable $D_e$ estimates up to 5 times the $D_0$ value of the DRC, which is far beyond the conventional limit of $2D_0$.

We first investigated if individual grains from each of the Enderby samples could be used to develop a SGC. Since the latter should be established using only grains that are considered well-behaved(Li et al., 2016) we applied the same rejection criteria as described above, except that grains with natural signals in saturation were accepted for purposes of establishing the SGC. The $L_x/T_x$ ratios of all of the accepted grains from each of the samples are plotted against the corresponding regenerative doses in Fig. S5a, which shows the large variation in $L_x/T_x$ ratios at any particular dose. We then applied the least-square normalisation (LS-normalisation) procedure of Li et al.(2016) to these data, resulting in significantly reduced scatter and a common SGC for the accepted grains from each of the samples (the best-fit curve in Fig. S5b). Next, the $L_n/T_n$ ratio for each grain was multiplied by a scaling factor determined from the LS-normalisation procedure (Li et al., 2016). The distributions of LS-normalised single-grain $L_n/T_n$ ratios are displayed in Fig. S6. Enderby-OSL1 (Fig. S6a) and -OSL2 (Fig. S6b) both appear to have two distinct populations of grains, with the vast majority of grains (75% in Enderby-OSL1 and 94% in Enderby-OSL2) comprising one component and the other component consisting of 2–3 grains with much smaller $L_n/T_n$ ratios; we interpret the latter as younger, intrusive grains. We applied the finite mixture model (FMM; (Roberts et al., 2000)) to both distributions determine the weighted-mean $L_n/T_n$ ratio for each component, assuming a two-component mixture. In the model, the $L_n/T_n$ overdispersion value was varied between 0 and 40%, and the lowest Bayes Information Criterion score was used to identify the optimum fit (David et al., 2007;Jacobs et al., 2011;Jacobs et al., 2008).

In contrast, Enderby-OSL3 (Fig. S6c) and -OSL4 (Fig. S6d) each appear to consist of a single population of grains, so we determined the weighted-mean $L_n/T_n$ ratio for these two distributions using

the central age model (CAM (Galbraith et al., 1999)). The weighted-mean ratio for Enderby-3 is based on just 4 grains, so the relative standard error on this estimate is correspondingly large (35%). The weighted mean $L_n/T_n$ ratios for Enderby-OSL3 and -OSL4, and for the major FMM components of Enderby-OSL1 and -OSL2, were then projected on to the SGC to estimate the corresponding $D_e$ values (Table 3).

The environmental dose rates were estimated from the sediment samples recovered from each tube hole. The total dose rate to K-feldspar grains consists of 4 components: the external gamma, beta and cosmic-ray dose rates, and the internal beta dose rate. The external gamma and beta dose rates were estimated from thick-source alpha counting (Aitken, 1985) and low-level beta counting (Jacobs and Roberts, 2015;Bøtter-Jensen and Mejdahl, 1988). The minor contribution from cosmic rays was estimated from the present-day burial depth of the samples and the latitude, longitude and altitude of the site (Prescott and Hutton, 1994). These external components of the total dose rate were adjusted for water content, assuming a long-term value of 25% (which falls within the range of measured water contents: 17–26%) and a standard error of ± 6% to capture any plausible variations in time-averaged water content over the entire period of sample burial. The internal beta dose rate was estimated by assuming K and Rb concentrations of 10 ± 2% and 400 ± 100 ppm, respectively (Huntley and Hancock, 2001;Smedley et al., 2012). The dosimetry data are summarised in Table 3.

## 2. Flowline model description

The glacier model employed is essentially the one-dimensional shallow-ice approximation (SIA) flowline model of Golledge & Levy (2011), with two modifications implemented for this study. First, the positive degree-day (PDD) scheme formerly used to calculate surface mass balance (SMB) has been replaced by a simplified energy-balance scheme, following the insolation-temperature melt method (Eq. 16 of Robinson et al., (2010)). This formulation allows for orbital variability to be included in the calculation of SMB by incorporating time-dependent insolation values from the solution of Laskar *et al.*, (2004). Shortwave radiation is specifically accounted for (unlike in the PDD scheme), since SMB is controlled in part by atmospheric transmissivity (the ratio between downward shortwave radiation at the

land surface and at the top of the atmosphere). Atmospheric transmissivity is modified from a base value according to surface elevation (Robinson et al., 2010). Spatial and temporal variations in surface albedo are included, with values ranging from 0.7 (snow) to 0.05 (ocean). A fractional albedo value is calculated to account for changes in surface type within each cell during the year. Resultant snowfall is prevented from accumulating on slopes steeper than 35° and instead is redistributed (progressively if necessary) to the next downslope cell.

The second modification to the model is the way in which basal sliding velocities are calculated. Previously we used a triangular averaging scheme (Kamb and Echelmeyer, 1986) to smooth calculated driving stress values, which were then used to calculate sliding velocities using a rate factor and sliding exponent (Eq. 8 of Golledge and Levy (2011)). Here we follow instead the approach of Bueler & Brown (2009), and superpose velocity solutions from the SIA with those from the shallow-shelf approximation (SSA). The latter essentially represents the sliding component of glacier flow, by explicitly incorporating the effects of longitudinal stresses. Our iterative scheme follows MacAyeal (1997): ice thickness and the horizontal velocity gradient are first used to calculate an effective viscosity, from which the horizontal strain rate is determined and then used to update the calculated sliding velocity. We use a coefficient to modify calculated sliding values to account for unknown basal drag.

## 3. OxCal code

```
Plot()
 {Curve("ShCal13","ShCal13.14c");
  Outlier_Model("General", T(5), U(0,4), t);
  Sequence()
 {Boundary("Start 1");
   Phase("1")
   {R_Date("Lower Lake Speight Terrace",
12306, 48)
    {Outlier("General", 0.05); };
   R_Date("O Transect", 12173, 67)
   {Outlier("General", 0.05);};
   R_Date("Ranui Cove", 9374, 28)

   {Outlier("General", 0.05);};
   R_Date("Sandy Bay 2", 13020, 36)
   {Outlier("General", 0.05);};
   R_Date("Deas Head", 11080, 120)
   {Outlier("General", 0.05);};
   R_Date("Tagua Bay", 9717, 30)
   {Outlier("General", 0.05);};
   R_Date("Red Tape Bog", 11951, 95)
   {Outlier("General", 0.05);};
   R_Date("Near O on transect", 9057, 52)
   {Outlier("General", 0.05) };
   R_Date("North Pt Bay", 9292, 98)
```

```
        {Outlier("General", 0.05);};
        R_Date("Mt Hooker", 10859, 77)
        {Outlier("General", 0.05);};
        R_Date("Hooker Cliffs", 12950, 200)
{Outlier("General", 0.05);};
        R_Date("Carnley Harbour 3", 10143, 30)
        {Outlier("General", 0.05);};
        R_Date("Rocky Bay", 11700, 140)
        {Outlier("General", 0.05);};
R_Date("Lower Lake Speight", 10150, 40)
        {Outlier("General", 0.05);};
        R_Date("Musgrave", 9222, 30)
        {Outlier("General", 0.05);};
        R_Date("Pillar Rock", 10086, 31)

{Outlier("General", 0.05);};
        R_Date("Carnley core 2", 10681, 29)
        {Outlier("General", 0.05);};
        R_Date("Homestead Bog Ridge", 12780,
    120)
{Outlier("General", 0.05);};
        R_Date("Homestead Scarp", 13648, 73)
        {Outlier("General", 0.05);};
        R_Date("Mt Honey Saddle", 12445, 76)
        {Outlier("General", 0.05);};
R_Date("Col Ridge", 9416, 57)
        {Outlier("General", 0.05);};};
      Boundary("End 1");};};
```

## Data availability

Data will be lodged on the NOAA Paleoclimate Archive

## Acknowledgements

First and foremost, we must acknowledge the contribution of Dr Richard Jones, who was integral to this work, and is dearly missed. The work was supported by the Australasian Antarctic Expedition 2013-

2014, the Australian Research Council (FL100100195, FT120100004, DE130101336 and DP130104156) and the University of New South Wales. MM and JW were supported by Core Funding for Crown Research Institutes, from the New Zealand Ministry of Business, Innovation and Employment's Science and Innovation Group. A major thanks to the captain and crew of the *MV Akademik Shokalskiy*, James MacDiarmid and Ben Fink, and Henk Haazen and Kali Kahn on the *Tiama*

for all their help in the field. Research on the New Zealand subantarctic Auckland and Campbell Islands was undertaken under the New Zealand Department of Conservation National Authorisation Numbers 37687-FAU and 39761-RES. Thanks to Jenny Dahl of the Rafter Radiocarbon Laboratory for advising on radiocarbon age NZA4509.

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

**Tables**

| # | Location | Site | Lat (S) | Long (E) | Elevation (m) | LTEL Lab code | Lab code | Depth (cm) | Material | d13C | Uncalibrated age (14C yrs BP) | +/- | Calibrated mean age (yrs BP) | +/- |
|---|---|---|---|---|---|---|---|---|---|---|---|---|---|---|
| 1 | Auckland | Sandy Bay 2 | 50°29'55.90"S | 166°17'23.80"E | 5 | X13/82 | Wk-38429 NZ | 119 | Peat | n.m | 13020 | 36 | 15502 | 206 |
| 2 | Auckland | North Point Bay | 50°31'0.09"S | 166°10'29.60"E | 40 | X94/16 | 8226 | 45 | Wood | -28.8 | 9292 | 98 | 10440 | 131 |
| 3 | Auckland | Pillar Rock Exposure | 50°30'58.29"S | 166°12'59.07"E | 20 | X14/52 | WK-40801 NZA | 119 | Peat | n.m. | 10086 | 31 | 11549 | 122 |
| 4 | Auckland | Dea's Head[1] | 50°31'28.57"S | 166°13'5.55"E | 25 | X94/12 | 4608 | 440 | Silty peat | -27 | 11080 | 120 | 12910 | 114 |
| 5 | Auckland | Red Tape Bog | 50°31'29.85"S | 166°12'59.73"E | 20 | X94/11 | NZA 4607 | 563-570 | Peat | -27.31 | 11951 | 95 | 13749 | 161 |
| 6 | Auckland | Near O Transect | 50°31'37.42"S | 166°12'42.60"E | 20 | X94/15 | WK 19756 | 105 | Bulk peat | -27.2 | 9057 | 52 | 10195 | 179 |
| 7 | Auckland | O Transect | 50°31'36.58"S | 166°12'42.08"E | 20 | X94/14 | WK 19734 | 340 | Bulk peat | -28.2 | 12173 | 67 | 13984 | 149 |
| 8 | Auckland | Ranui Cove[2] | 50°32'12.60"S | 166°15'48.48"E | 20 | X13/106 | Wk-38424 NZA | 651 | Peat | n.m. | 9374 | 28 | 10547 | 135 |
| 9 | Auckland | Mt Hooker | 50°32'51.85"S | 166°10'42.42"E | 275 | X94/17 | 9293 | 185 | Peat | -27.9 | 10859 | 77 | 12732 | 64 |
| 10 | Auckland | Musgrave Harbour | 50°47'13.65"S | 165°58'18.71"E | 52 | X14/50 | Wk-40795 | 175 | Peat | n.m. | 9222 | 30 | 10338 | 62 |
| 11 | Auckland | Carnley Harbour 2 | 50°48'12.47"S | 166° 0'43.79"E | n/a | n/a | Wk-39916 | 29-30 | Peat | n.m. | 10681 | 29 | 12619 | 53 |
| 12 | Auckland | Carnley Harbour 3 | 50°49'39.47"S | 166° 0'41.74"E | n/a | n/a | Wk-39919 | 49.5-50 | Peat | n.m | 10143 | 30 | 11686 | 126 |
| 13 | Auckland | Tagua Bay | 50°48'54.36"S | 166° 3'47.31"E | 157 | X13/87 | Wk-38425 | 288 | Peat | n.m. | 9717 | 30 | 11071 | 117 |
| 14 | Auckland | Lower Lake Speight Corrie | 50°49'34.90"S | 165°59'14.06"E | 189 | X14/50 | WK 41406 | 108 | Silty peat | n.m. | 10150 | 40 | 11698 | 162 |
| 15 | Auckland | Lower Lake Speight Terrace | 50°49'31.78"S | 165°59'46.78"E | 120 | X13/85 | WK 39079 | 186 | Peat | n.m. | 12306 | 48 | 14199 | 154 |
| 16 | Campbell | Hooker Cliffs[3] | 52°28'19.16"S | 169°11'36.08"E | 60 | M84/11 | NZ6898 | 440 | Peat | n.m. | 12950 | 200 | 15400 | 364 |
| 17 | Campbell | Col Ridge Crest | 52°32'20.23"S | 169° 7'37.90"E | 200 | X99/9 | WK 14730 | 157 | Peat | -29.6 | 9416 | 57 | 10599 | 104 |
| 18 | Campbell | Homestead Scarp[4] | 52°32'55.83"S | 169° 8'7.56"E | 30 | X99/11 | WK 19746 | 390 | Bulk peat Wood | -29.2 | 13648 | 73 | 16286 | 332 |

**Table 1:** Radiocarbon dating of basal peat from sediment cores across Campbell and Auckland Islands used to model peat growth onset. Column 11: n.m. marks samples where $\delta^{13}C$ not measured owing to small sample size. Core numbers align with those on Fig. 1. Previously published dates shown by superscript numbers: [1](McGlone et al., 2000); [2](McGlone, 2002); [3](McGlone et al., 1997); [4](McGlone et al., 2010).

**Table 2:** Radiocarbon dating for sedimentary sequences taken from Pillar Rock, Musgrave Harbour, and Lower 'Lake' Speight cirque and terrace.

| Sediment sequence | Lab code | Depth (cm) | Material | Uncalibrated age (14C yrs BP) | +/- | Calibrated mean age (yrs BP) | +/- |
|---|---|---|---|---|---|---|---|
| LLS Corrie | Wk-41404 | 94 | Peat | 8359 | 36 | 9330 | 70 |
| LLS Corrie | Wk-41405 | 100 | Peat | 8182 | 34 | 9090 | 60 |
| LLS Corrie | Wk-41406 | 108 | Mineral peat | 10150 | 40 | 11699 | 135 |
| LLS Corrie | Wk-41407 | 149 | Silty peat | 5630 | 26 | n/a | n/a |
| LLS Corrie | Wk-41408 | 153.5 | Silty peat | 7449 | 40 | n/a | n/a |
| | | | | | | | |
| LLS Terrace | Wk-39076 | 50 | Peat | 5445 | 25 | 6220 | 50 |
| LLS Terrace | Wk-39077 | 99 | Peat | 7279 | 28 | 8060 | 50 |
| LLS Terrace | Wk-39078 | 170 | Peat | 11952 | 36 | 13710 | 70 |
| LLS Terrace | Wk-41149 | 175 | Peat | 12225 | 44 | 14070 | 80 |
| LLS Terrace | Wk-39079 | 185 | Peat | 12306 | 48 | 14200 | 122 |
| LLS Terrace | Wk-41150 | 196 | Sandy peat | 11729 | 36 | n/a | n/a |
| LLS Terrace | Wk-38423 | 209 | Sandy peat | 11409 | 31 | n/a | n/a |
| | | | | | | | |
| Pillar Rock | Wk-40800 | 94 | Peat | 8106 | 27 | 8960 | 80 |
| Pillar Rock | Wk-40801 | 119 | Peat | 10086 | 31 | 11549 | 110 |
| Pillar Rock | Wk-40799 | 238 | Peat | 36858 | 597 | 41350 | 500 |
| Pillar Rock | Wk-40802 | 248 | Peat | 42234 | 203 | 45900 | 1250 |

## Figures

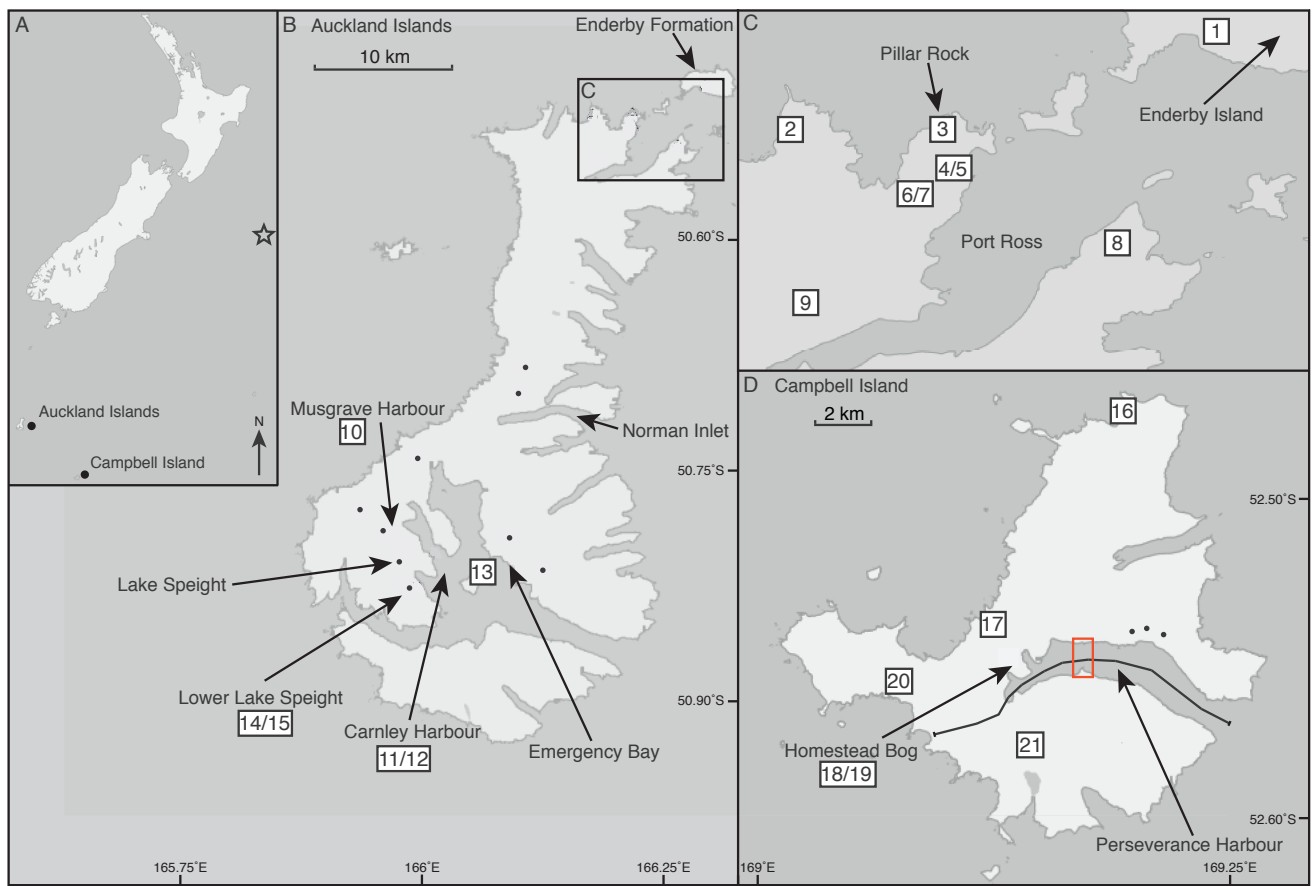

**Figure 1:** Map of Auckland and Campbell Islands. (A) Inset showing location relative to New Zealand mainland. Star shows location of ocean core DSDP 594 used in flowline modelling; (B) Auckland Islands. Black box marks location of inset C; (C) Inset showing northeast Auckland Islands; (D) Campbell Island. Model flowline marked along Perseverance Harbour, red rectangle highlights location of submerged moraines. Locations of basal peat radiocarbon ($^{14}$C) dates given by core numbers (see main text and Table 1). Key glacial cirques marked by black dots.

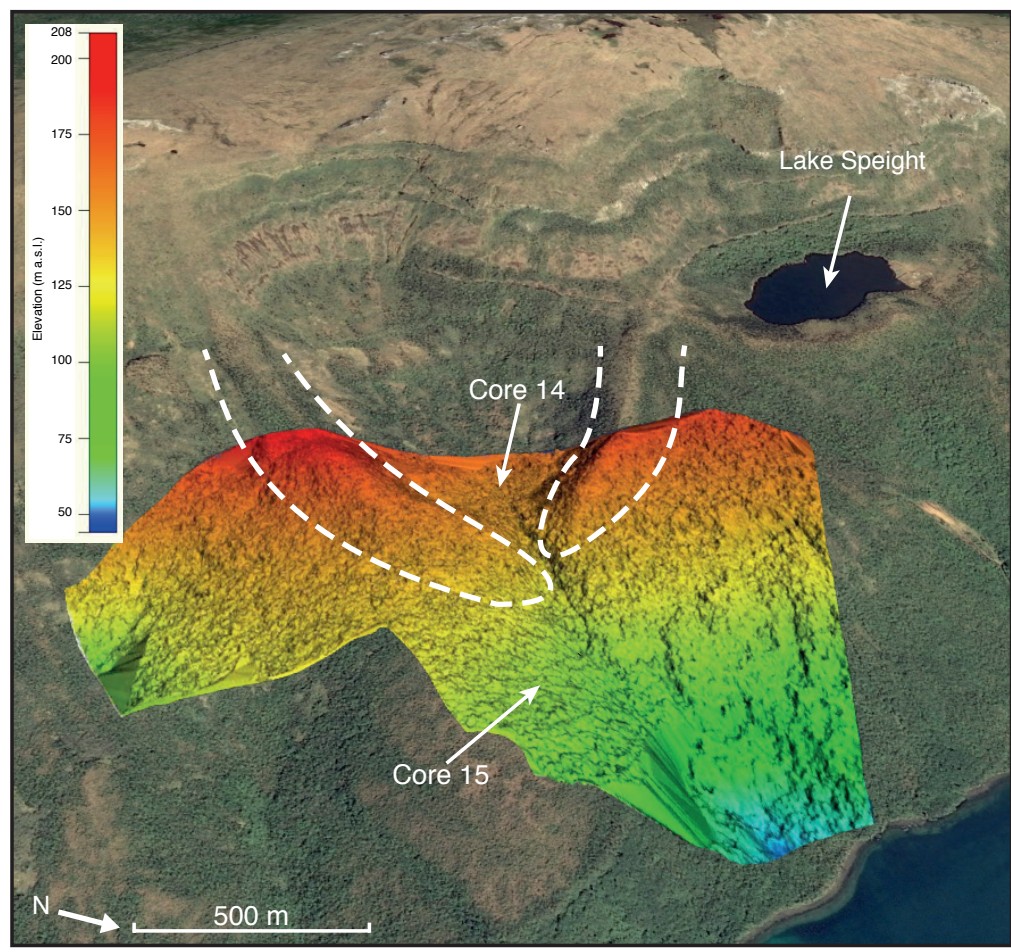

**Figure 2:** Digital elevation model of Lower Lake Speight Cirque, captured by Unmanned Aerial Vehicle, overlain on imagery from Google Earth. Moraines are highlighted by dashed white lines, with locations of Cores 14 and 15 given by arrows.

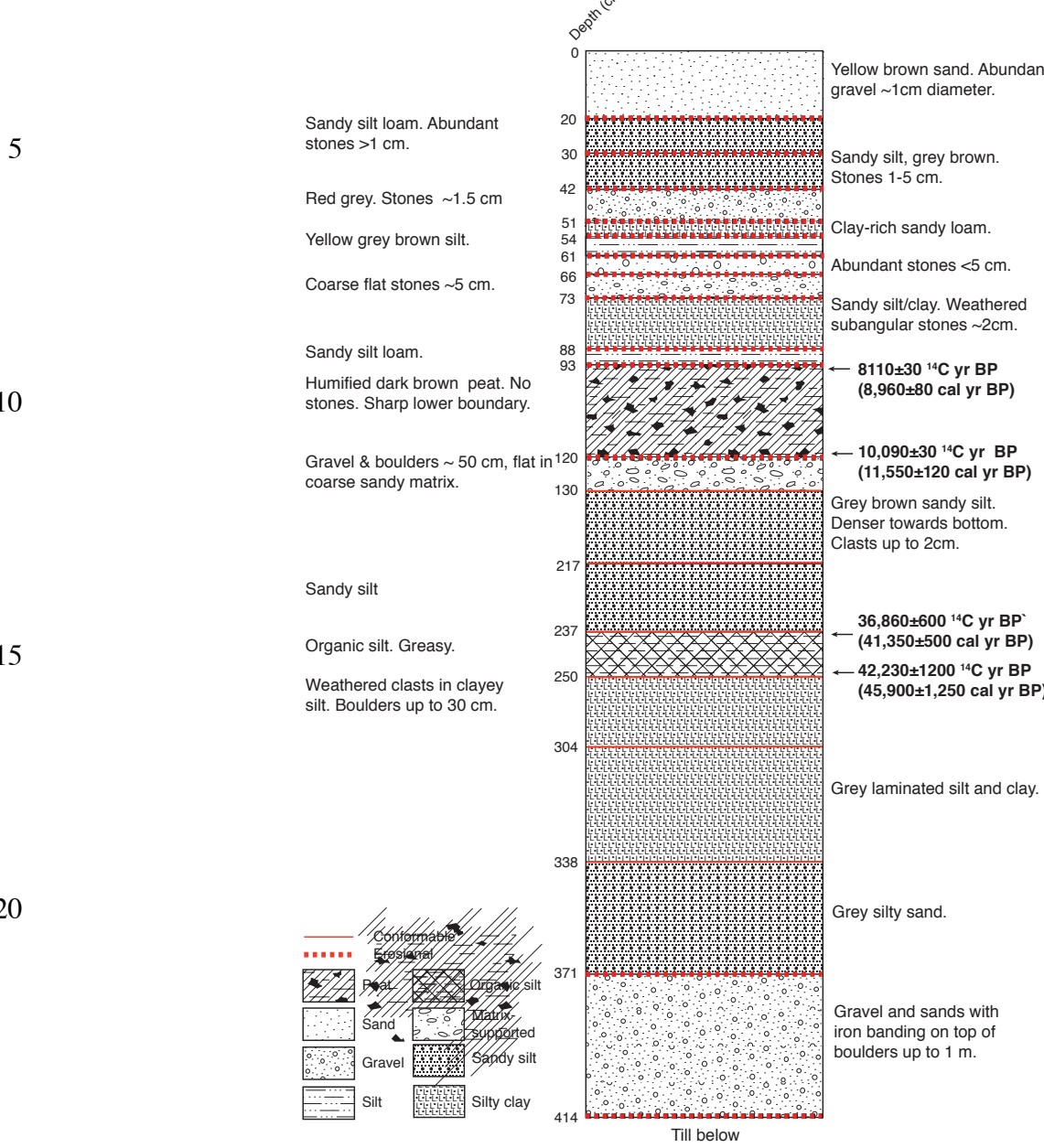

**Figure 3:** Schematic of exposed sediment section surveyed and sampled at Pillar Rock, Auckland Island. Radiocarbon dates of organic sections given as uncalibrated [14]C dates and calibrated mean ages. Not to scale.

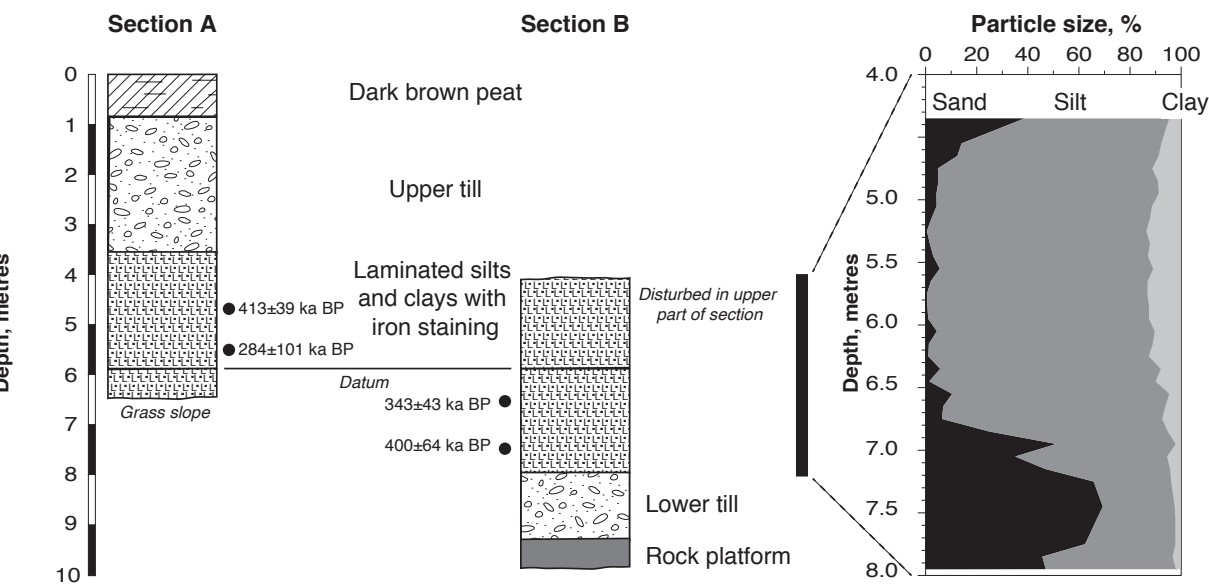

**Figure 4:** Schematic of exposed 'Enderby Formation' on Enderby Island. Optical dating sample locations shown by black circles ('Enderby-1' at bottom, 'Enderby-4' at top). Particle size analysis of laminated silts blown up at right. Sections A and B are separated by an eroded 'step', but have multiple prominent lithostratigraphic layers which extend from one section to the other (see Figure S7B).

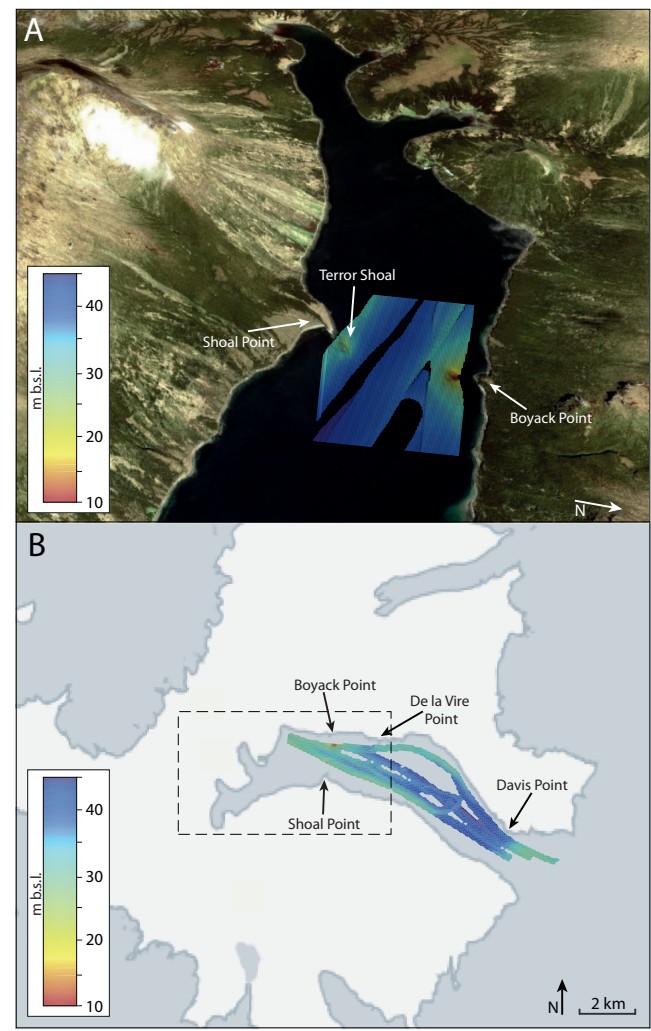

**Figure 5:** Multibeam surveys of Perseverance Harbour, Campbell Island (A) Multibeam imagery of sub-marine moraines at Boyack Point and Shoal Point, respectively the north and south terrestrial expressions of the moraines, with Terror Shoal long known as a navigational hazard (Ross, 1847); (B) Multibeam imagery along the length of Perseverance Harbour. No other submerged moraine features were found, including at De la Vire and Davis Points, which could be considered similar terrestrial expressions.

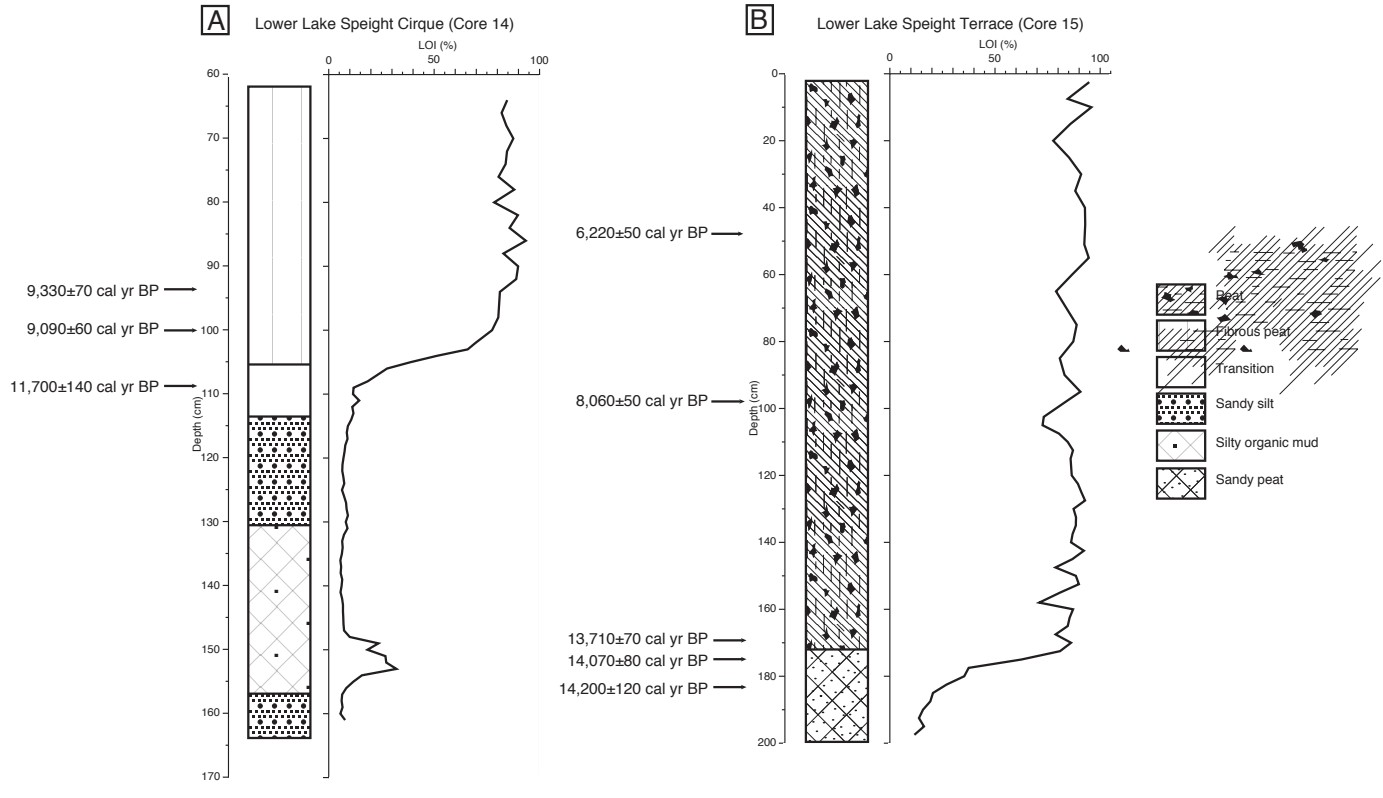

**Figure 6:** Schematic of sediment cores taken from Lower Lake Speight Cirque and Terrace. Radiocarbon dates given as calibrated ages. Loss-on-ignition (LOI; %) curve shown alongside cores.

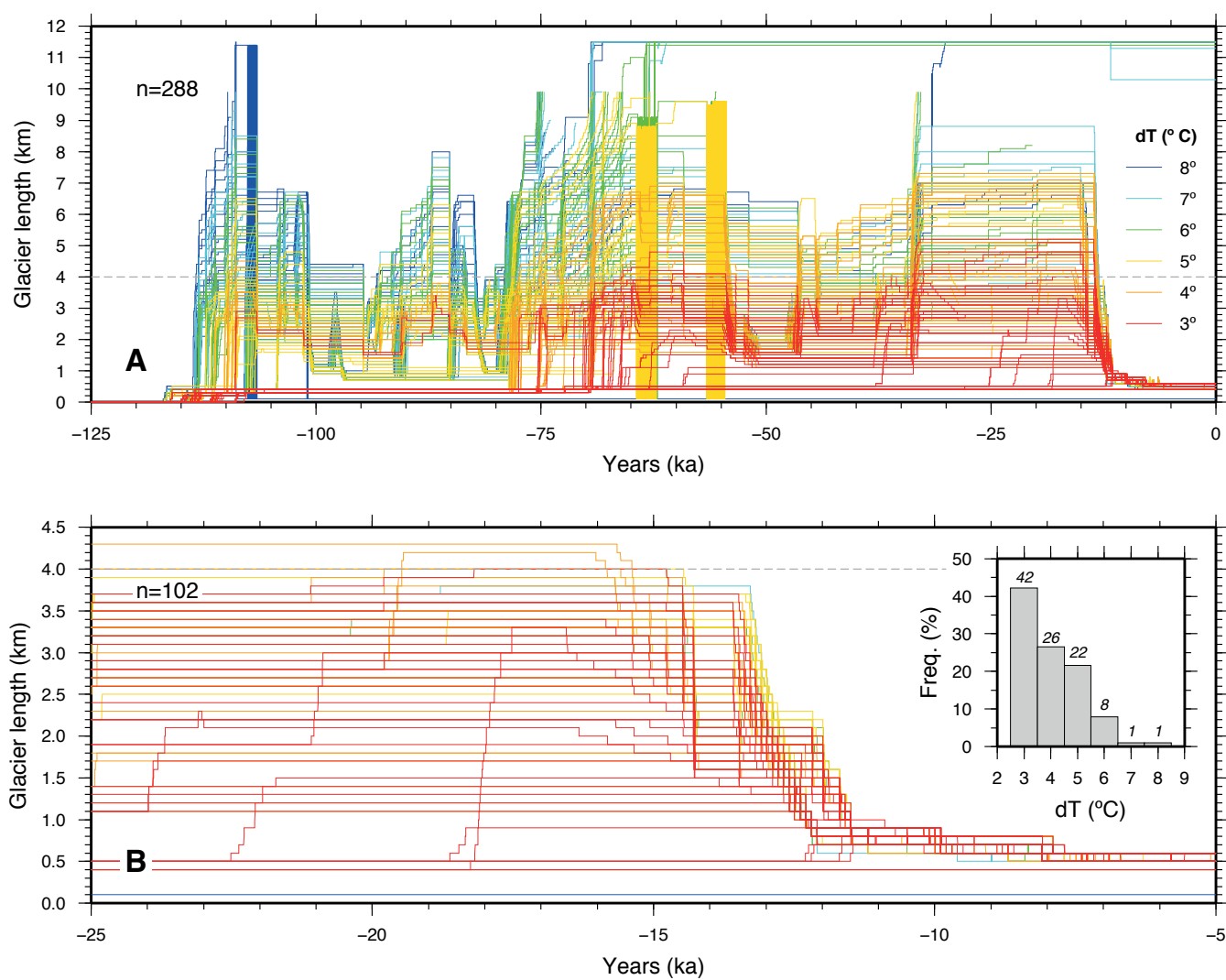

**Figure 7:** Ensemble of glacier simulations, coloured according to the mean annual air temperature depression used to scale the EPICA Dome C or sea surface temperature timeseries forcing of (A) all 288 flowline model runs and (B) the 102 simulations that fit the radiocarbon constraint. Inset shows the frequency of MAAT depressions in this subset from 3-8°C.

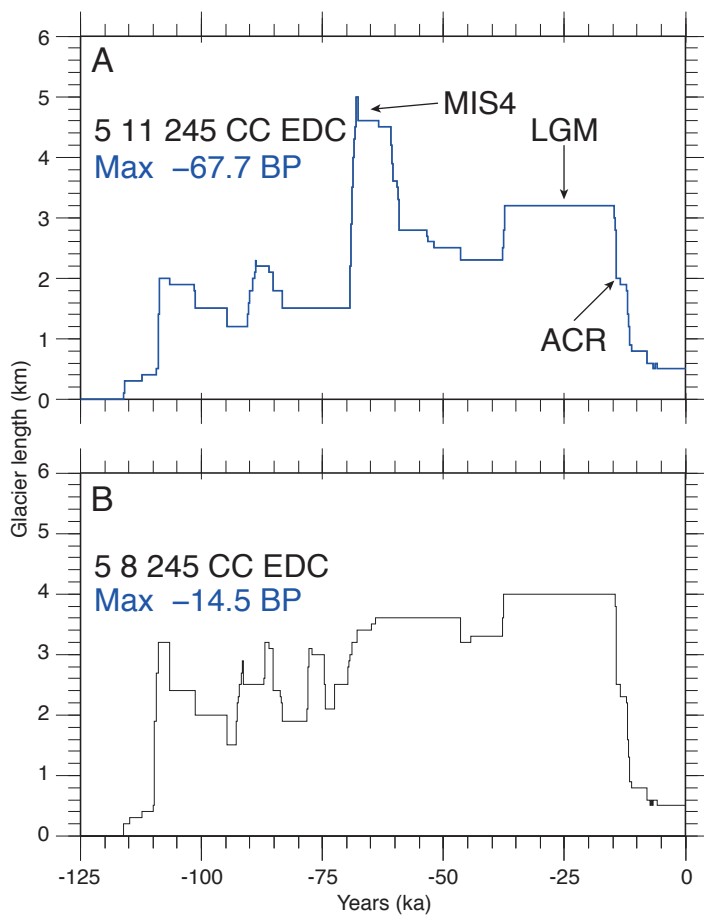

**Figure 8:** (A) Model output showing best fit of glacial extent to palaeoclimatic and geomorphological data, with mean annual air temperature depression of 5°C, seasonality range of 11°C, temperature-coupled precipitation regime and forced using the EPICA Dome C temperature data. Marine Isotope Stage 4, Last Glacial Maximum and Antarctic Cold Reversal are labelled. (B) Model output with identical climatic parameters to (A), except seasonality which is 8°C.

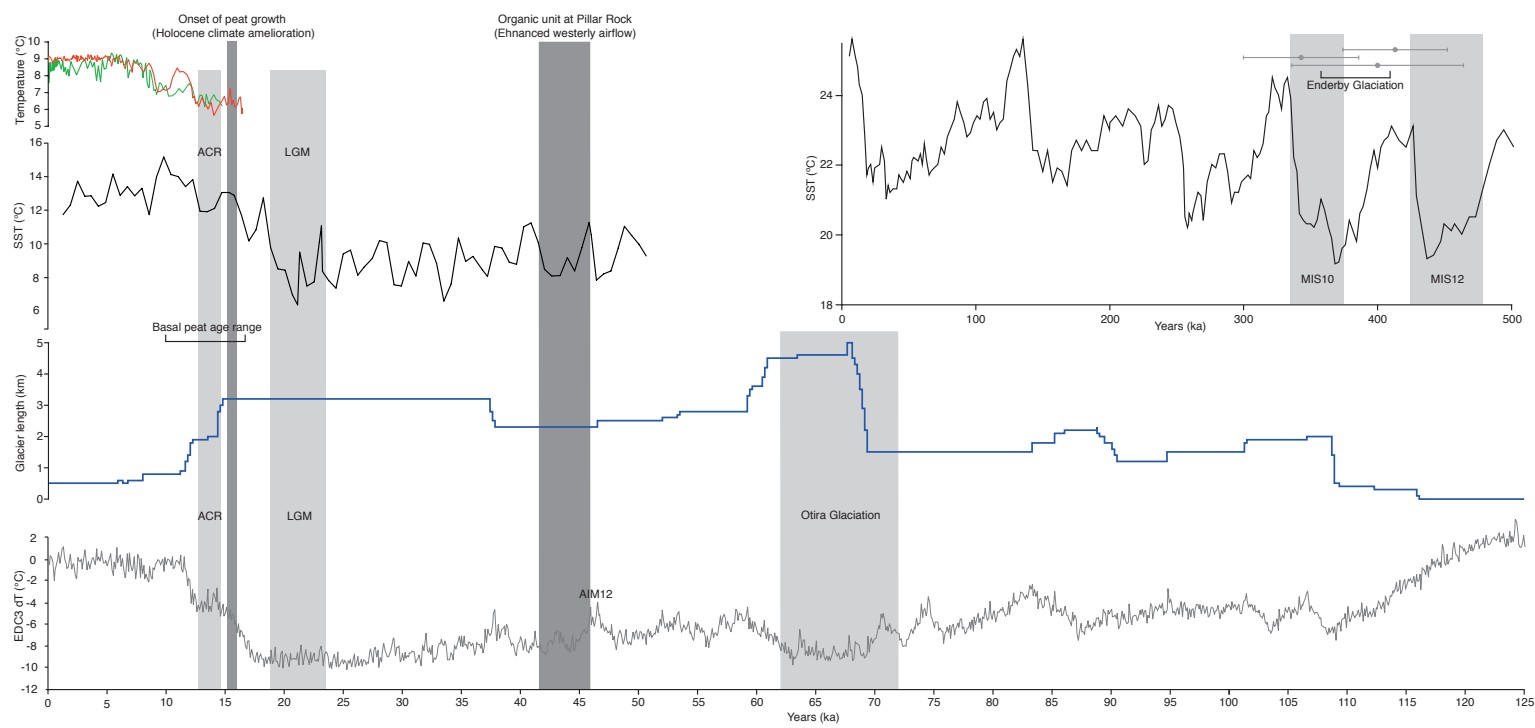

**Figure 9:** Plot showing key Late Pleistocene events on the islands. Organic growth at ~46-41 ka and onset of peat growth at 16.6 ka are shown by dark grey bars against sea surface temperature reconstructions (Pahnke and Sachs, 2006) and EPICA Dome C δT (Parrenin et al., 2007). The ACR and New Zealand timing of the LGM are highlighted by pale grey bars. Summer temperature reconstructions from Campbell Island (McGlone et al., 2010) are shown in green (Mount Honey) and red (Homestead Scarp), highlighting minimal ACR temperature depression over the islands despite the