# Peer review of "Pleistocene glacial history of the New Zealand subantarctic islands"

_Climate of the Past, 2018_

## Referee Comment (RC1) · Anonymous Referee #1 · 25 Jun 2018

This paper "Pleistocene glacial history of the New Zealand subantarctic islands" by E. Rainsley et al. presents new data, from the subantarctic Islands Auckland and Campbell, and a flowline modelling of a large U-shaped valley if the Campbell Island. The paper is well written and the results concerning the flowline simulations implying a large seasonal temperature amplitude during the last glacial maxim highlights the necessity of reconstructing not only mean annual temperature trends but also the seasonal amplitude. While this paper deserves publication in Climate of the Past, I have some concerns regarding the lack of modern reference for other subantarctic Islands glaciation, local subtropical front shifts, westerlies migration. For exemple, the changes in position of the Subtropical front in this area are detailed in Bostock et al. 2015 and Bostock et

al. 2013 presents a review of the climate changes in the Australian-New Zealand sector of the Southern Ocean. Jomelli et al., 2017 a, 2018, discuss the evolution of glacier on the subantarctic Kerguelen Archipelago since 50kyrs and Boex et al., and others, in Patagonia. The role of the Westerlies latitudinal migration for glacier evolution is discussed but with no reference to the different published papers (in Patagonia: Moreno et al, Montade et al., Lamy et al.,..., in Kerguelen: Van der Putten all, 2015...) Other minor comments/corrections are listed: Chronology: would it be possible to date (14C) the peat cores with plant remains instead of bulk organic matter, to increase the robustness of the age model? Concerning the Enderby formation, I'm confused: page 4 the authors explain that pollen and spores are present in the laminated silt that has been sampled and that separate the two layers of glacial till. On page 7, 2.3.2 the authors explain that no organic material was present to provide an age constraint for the Enderby Formation. Concerning the dating results, I wonder how the authors obtained an onset of peat growth at ∼17,23-16,11 cal age with the Oxcal Phase age calibration as, in Table 1, dating of basal peat, the oldest cal age is 15,47 cal age. It would be nice to indicate the oxcal parameters chosen in the supplementary material. Concerning the Enderby formation, I suppose the authors did a weighted average of the different IRSL ages, but then they should get 378±26 ky ? As those dates are from the laminated silt in between the two glacial till, could the glacial till be from the isotopic stage 12 and 10 respectively? Page 12, line 25 correct Table 1 instead of Table 2. As the sub-tropical front is considered to have shifted latitudinally during the last glacial cycle, is it really robust to consider the temperature difference between Auckland and Campbell Islands and EDC or ODP584 constant? Page 14, Line 1-2, replace "Fig.6 and Fig.S8" by Fig. 8 Page 15, line 5, Classically Mid-Pleistocene is associated with the Mid-Pleistocene transition (∼1.3 to 0.9 Ma) Maybe it would be easier to call it the Mid Bruhnes?

---

## Referee Comment (RC2) · Anonymous Referee #2 · 9 Aug 2018

The manuscript by Eleanor Rainsley and co-authors presents a wealth of new data from two subantarcitc islands south of New Zealand, which will revise the current understanding of the late Quaternary glacial history in this remote area at high southern latitude. The authors applied a multi-disciplinary approach using different archives and synthesized this proxy data with glacier flow modeling. The science behind the study is solid and the reconstructed glacial history clearly deserves publication in the Journal Climate of the Past. However, the manuscript lacks in some parts a clear structure and needs some modifications.

1. Structure and content of the methods and results chapter

- In chapter 2, the authors often talk not about the methods, but about study locations and even present results in this chapter. Chapter 2.2 is entitled "Sedimentary analysis",

but the authors only discuss the different study sites. The same is true for a large part of chapter 2.3 about the chronology. There is very little information about the dating methods, but mostly explaining the different study sites and what was dated. The last part of chapter 2.3.2 contains even the results of the optical dating approach. So maybe the paper needs a chapter, where the authors introduce the different study sites in a more systematic way. That would make it easier for the reader to follow the different arguments of the authors.

- In the first sections in the chapter 3.1 "Geomorphology" the authors discuss the results of Enderby Till analysis. . ..but this has nothing to do with geomorphology. Therefore, I would not discuss it under the title "geomorphology".

- In addition, the authors decided to have an appendix with extended information about the methodology. However, there is a large overlap between chapter 2 and the appendix for the geomorphological survey and the sediment analysis. In my view, I would include the information in the extended methodology in the main text for these two sections. The extended methodology for the chronology and the flowline model description makes sense.

2. Basal peat ages (p. 12, line 15ff)

It would be nice to have more information about these basal ages. Why not plotting these basal ages against a time axis so that the reader can see distribution of the onsets of peat growth. Maybe include this data in a synthesis figure (see comment 4).

3. Perseverance Harbour moraine

In Figure 5, the authors show multibeam data that potentially indicate the presence of moraine ridge on the floor of the inlet. The question would then be if the ridge at Shoal Point is also of glacial origin. Looking at Google Earth, Shoal Point is a limited, but very straight structure, which I rather attribute as a 'hardrock' feature. Are there any indications that the Shoal Point has any glacial deposit on land? See also comment for

Fig. S9.

4. Synthesis figure

What I miss in this paper is a figure that synthesizes the (many!) results of this study. Figure 9 currently only shows two gray bars (!) representing a tiny amount of the generated results! Why not compiling your data including all the dates from the onset of the peat formation, the modeled glacial length, and so on. Maybe extend the time axis further back in time (evt. with axis breaks). That would greatly increase the impact of the paper.

Minor remarks:

Abstract: The maximum ice extent around 68 ka is not mentioned in the abstract.

p. 6, line 18: Explain why the top part of the LLS Cirque core was not sampled between 64-104: Lost? No recovery?

p.6, line 24: Mention here that the Enderby Formation was sampled at Site 1.

p. 9, line 2: Introduce the acronym NIWA

p.12, line 3: This should be Fig. S7D (and not S3D)

p. 12, line 27: . . .the age inversion is further UP (not down) the core. . .

p. 14, line 1: The subset of 25 simulations is shown in Fig. S10!

p. 14, line 22: Port Ross is not labeled on the map in Fig 1.

p. 19, line 6ff: Here, the authors talk about the loss of catchment. Can this loss of catchment be quantified in order to judge if that is an important factor for the more recent glacials. For me it is hard to believe that this substantially modified the growth of the glaciers in the recent past.

p. 25, line 2: Space before reference

Figure 1: Site 20 is not mentioned in the manuscript

Figure 3: Maybe add on the side the extent of Fig. S7D for reference. Figure needs a higher resolution. Erosional contacts are very hard to see.

Figure 4: Short explain how the two sections were correlated.

Figure 9: Label in the figure should most likely be NZ eLGM (instead of NZ gLGM) In the figure caption: Space before reference (twice).

Figure S7C: Please explain what you want to highlight with the dashed red line/box and the red star.

Figure S9: In the hydrographic chart in Fig. S9A the position and the shape of the proposed moraine ridge in Norman Inlet is clear, but are there any signs of a moraine ridge visible on land in Fig. S9B? If so, then please mark the geomorphological features attributed to a glacial deposit in the photograph. Please give reference of the hydrographic chart.

―――――――――――――――――――――

---

## Author Response (AR1)

**Reviewer 1**

Many thanks to Reviewer 1 for their review, and their recognition of the importance of our study's results. We respond to their suggestions and queries below.

I have some concerns regarding the lack of modern reference for other subantarctic Islands glaciation, local subtropical front shifts, westerlies migration. For example, the changes in position of the Subtropical front in this area are detailed in Bostock et al. 2015 and Bostock et al. 2013 presents a review of the climate changes in the Australian-New Zealand sector of the Southern Ocean.

We thank the reviewer for highlighting these papers, and will certainly include them within our revised manuscript. Bostock et al 2015, for example, presents evidence that the subtropical front shifted may have actually shifted south in the Last Glacial maximum, in contrast to the strong northwards shift during MIS-10 and -12. This provides further explanation for the severely limited LGM glaciation we found in the New Zealand subantarctics when compared to the extensive ice cap conditions experienced in MIS10/12.

Jomelli et al., 2017 a, 2018, discuss the evolution of glacier on the subantarctic Kerguelen Archipelago since 50kyrs and Boex et al., and others, in Patagonia. The role of the Westerlies latitudinal migration for glacier evolution is discussed but with no reference to the different published papers (in Patagonia: Moreno et al, Montade et al., Lamy et al.,..., in Kerguelen: Van der Putten all, 2015: : :)

We agree with the reviewer that there are important connections between the results we present here and the behaviour of glacier evolution in other parts of the southern hemisphere. We discuss this throughout our paper (i.e. pages 3, 9, 15, 17 and 21), but will include these extra references relevant to the wider discussion (without overly complicating what is already a somewhat lengthy manuscript!). The recently published studies by Jomelli et al are particularly relevant, as they find strong evidence for a decline in glacial extent in the subantarctic Kergulen islands throughout the LGM, further highlighting the presence of a mechanism that restricted LGM glacial expansion on a potentially hemispheric level.

Other minor comments/corrections are listed: Chronology: would it be possible to date (14C) the peat cores with plant remains instead of bulk organic matter, to increase the robustness of the age model?

The peat in the subantarctics core samples is highly humified with little in the way of preserved macrofossils or seeds, hence the bulk peat dates. This humification is particularly profound in the old Pillar Rock samples; the peat found here is so well decomposed that it has an almost waxy texture. We thank the reviewer for seeking clarification and we will include an explanation as to why bulk peat samples were chosen in our revised methods section. We consider that the number of cores from which we have sourced basal dates, together with their consistent age models, and the fact that many papers from the region have been published with excellent age depth models constructed from bulk peat dates, means that our dating is robust.

Concerning the Enderby formation, I'm confused: page 4 the authors explain that pollen and spores are present in the laminated silt that has been sampled and that separate the two layers of glacial till. On page 7, 2.3.2 the authors explain that no organic material was present to provide an age constraint for the Enderby Formation.

We apologise for the confusion; the text on page 4 refers to work carried out by Fleming et al, published in 1976. This work found small quantities of pollen in four samples from the laminated sediments found between the two layers of glacial till, and interpreted this to mean that the tills came from separate glaciations. Conversely, we comprehensively sampled the laminated silt at 2 cm intervals, and found no pollen or spores in any of the samples. We shall make this clearer in our revised manuscript.

Concerning the dating results, I wonder how the authors obtained an onset of peat growth at \_17,23-16,11 cal age with the Oxcal Phase age calibration as, in Table 1, dating of basal peat, the oldest cal age is 15,47 cal age. It would be nice to indicate the oxcal parameters chosen in the supplementary material.

We apologise for the confusion. The reported ages were incorrectly given in the table. The correct age range for the initiation of peat growth is 16.8-16.3 ka cal BP (at 1 sd) with the oldest age obtained from Homestead Scarp of 16.3 $\pm$ 0.3 ka cal BP (at 1 sd). We will revise the table correctly and provide the OxCal code in the Supplementary Information file.

Concerning the Enderby formation, I suppose the authors did a weighted average of the different IRSL ages, but then they should get 378\_26 ky ?

When calculating the weighted mean of the IRSL samples, we excluded sample 'Enderby-3' as an outlier. We shall make this clear in our revised methods. As the reviewer has calculated, inclusion of this sample only alters the weighted mean by 6kyr, so in either case this does not affect our interpretation.

As those dates are from the laminated silt in between the two glacial till, could the glacial till be from the isotopic stage 12 and 10 respectively?

This is a possibility, but as we discuss on page 14, we interpret the conformable boundaries between the both the glacial diamictons and the marinated silt stone to suggest that there was not an extended period of time between deposition of the different parts of this sequence. This would also explain the lack of pollen and spores found in the laminated silts, which we would otherwise expect if it had been exposed for a period of several thousands of years.

Page 12, line 25 correct Table 1 instead of Table 2.

We thank the reviewer for picking this up, and will correct in our revised manuscript.

As the sub-tropical front is considered to have shifted latitudinally during the last glacial cycle, is it really robust to consider the temperature difference between Auckland and Campbell Islands and EDC or ODP584 constant?

This is an interesting point. We consider the proximity of the islands makes this unlikely but we will explicitly state this as an assumption. Thank you.

Page 14, Line 1-2, replace "Fig.6 and Fig.S8" by Fig. 8

Thanks to the reviewer for picking this up – this should read 'Fig. 8 and Fig. S10'

Page 15, line 5, Classically Mid-Pleistocene is associated with the Mid-Pleistocene transition (\_1.3 to 0.9 Ma) Maybe it would be easier to call it the Mid Bruhnes?

We thank the reviewer for the suggestion, but believe that the introduction of the term 'Mid Bruhnes' would cause greater confusion, as we do not link the findings in our study to the Mid Bruhnes Event. To avoid any misunderstandings, we will change this to 'middle of the Pleistocene' instead of 'Middle Pleistocene' in our revised manuscript.

**Reviewer 2**

We thank Reviewer 2 for their detailed and complimentary review, which highlights the value of our study to Quaternary science. We address their suggestions and queries below.

1. Structure and content of the methods and results chapter

The reviewer makes some helpful and sensible suggestions for improving the clarity and readability of this complex multidisciplinary paper, which we will act on in our revised manuscript.

2. Basal peat ages (p. 12, line 15ff)

Our constraint for the onset of peat growth, modelled in OxCal, is included as a bar in Figure 9. We will also provide a Kernel Density (KDE) distribution plot summarising the ages on the same figure (Bronk Ramsey, 2017).

**3. Perseverance Harbour moraine**

In Figure 5, the authors show multibeam data that potentially indicate the presence of moraine ridge on the floor of the inlet. The question would then be if the ridge at Shoal Point is also of glacial origin. Looking at Google Earth, Shoal Point is a limited, but very straight structure, which I rather attribute as a 'hardrock' feature. Are there any indications that the Shoal Point has any glacial deposit on land? See also comment for Fig. S9.

This is an important point to clarify. Unfortunately the large amount of peat covering the island prevented us from confirming the presence of glacial deposits. However, the mirror image features on either side of Perseverance Harbour (including the extension of these features up the valley side) argues strongly these are glacial in origin.

**4. Synthesis figure**

What I miss in this paper is a figure that synthesizes the (many!) results of this study. Figure 9 currently only shows two gray bars (!) representing a tiny amount of the generated results! Why not compiling your data including all the dates from the onset of the peat formation, the modeled glacial length, and so on. Maybe extend the time axis further back in time (evt. with axis breaks). That would greatly increase the impact of the paper.

Thanks to the reviewer for some suggestions of how to improve this figure and incorporate more of our results into it. We shall take these on board in our revised manuscript, and include the Kernel Distribution (KDE) of the 14C ages, the IRSL ages (with a break in the time scale), as well as our modelled glacier lengths. Set against the SST (subtropical front) record from Bard & Rickaby 2009, this will substantially improve the communication of our study's key results. We thank the reviewer for their excellent suggestion.

Minor remarks:

Abstract: The maximum ice extent around 68 ka is not mentioned in the abstract.

We have tried to simplify the wide-ranging findings of this multidisciplinary paper as much as possible within the abstract, but will add that our modelling, combined with field evidence, suggests a possible larger (than LGM, but smaller than the 384 ka maximum) glaciation at 68 ka.

p. 6, line 18: Explain why the top part of the LLS Cirque core was not sampled between 64-104: Lost? No recovery?

The focus of this study was the timing and impact of deglaciation. As such we did not undertake work on the Holocene part of the sequence. We will make this explicit in the revised text.

p.6, line 24: Mention here that the Enderby Formation was sampled at Site 1.

The Enderby Formation is located on the north coast of Enderby Island, as shown on Figure 1, distinct from Core Site 1 to the south of the island. We will clarify this in our revised manuscript.

p. 9, line 2: Introduce the acronym NIWA

NIWA is the National Institute of Water and Atmospheric Research, a Crown Research Institute of New Zealand. We will rectify this in our revised manuscript.

p.12, line 3: This should be Fig. S7D (and not S3D)

We thank the reviewer for spotting this.

p. 12, line 27 the age inversion is further UP (not down) the core

This is correct, thanks to Reviewer 2 for the correction.

p. 14, line 1: The subset of 25 simulations is shown in Fig. S10!

This will be corrected.

p. 14, line 22: Port Ross is not labeled on the map in Fig 1.

Our thanks to the reviewer for spotting this. We will rectify this in the revised manuscript.

p. 19, line 6: Here, the authors talk about the loss of catchment. Can this loss of catchment be quantified in order to judge if that is an important factor for the more recent glacials. For me it is hard to believe that this substantially modified the growth of the glaciers in the recent past.

The past erosion of basaltic rocky shorelines such as those found in the Auckland Islands is hard to quantify, further complicated by the oscillating global and regional sea levels over the past ~400 kyr covered by this study. Quantifying the possible effects of this erosion, as well as other potential contributors to the extensive MIS10 ice cap is beyond the bounds of our work here, but could form the basis for an interesting future investigation.

p. 25, line 2: Space before reference

This will be corrected.

Figure 1: Site 20 is not mentioned in the manuscript

A number of the sediment core sites are not discussed explicitly within the text of the manuscript, as they provided no data to the study beyond their basal peat dates. All such sites, including Site 20, are included in Table 1 and their location shown in Figure 1. Figure 3: Maybe add on the side the extent of Fig. S7D for reference. Figure needs a higher resolution. Erosional contacts are very hard to see.

We apologise for the low resolution of this figure in the discussion paper, which is an artefact of the submission guidelines that ask for figures to be included within the manuscript file. When uploaded separately for publication, the figures will be of their original high resolution, and much easier to view.

Figure 4: Short explain how the two sections were correlated.

The sequence has multiple prominent lithostratigraphic layers which extend from one section to the other. It was from one of these that we correlated the two sections. We will make this explicit in the revised version of the manuscript.

Figure 9: Label in the figure should most likely be NZ eLGM (instead of NZ gLGM) In the figure caption: Space before reference (twice).

To clarify, on this figure we have highlighted the global Last Glacial Maximum as defined by its timings in New Zealand, as we consider this the most relevant timeframe when considering the New Zealand subantarctics LGM extent.

Figure S7C: Please explain what you want to highlight with the dashed red line/box and the red star.

We thank the reviewer for picking up on this omission – we will include this information in the figure caption in the revised manuscript.

Figure S9: In the hydrographic chart in Fig. S9A the position and the shape of the proposed moraine ridge in Norman Inlet is clear, but are there any signs of a moraine ridge visible on land in Fig. S9B? If so, then please mark the geomorphological features attributed to a glacial deposit in the photograph. Please give reference of the hydrographic chart.

We thank the reviewer for spotting this and will add to the figure.

**Pleistocene glacial history of the New Zealand subantarctic islands**

Rainsley, Eleanor1\*, Turney, Chris S.M.2,3, Golledge, Nicholas R.4,5, Wilmshurst, Janet M.6,7, McGlone, Matt S.6, Hogg, Alan G.8,9, Li, Bo10,11, Thomas, Zoë A.2,3, Roberts, Richard10,11, Jones, Richard T.12,†, Palmer, Jonathan G.2,3, Flett, Verity13 de Wet, Gregory14, Hutchinson, David K.15, Lipson, Mathew J.2,

[revised manuscript text omitted]

ER 23/10/18 12:54 PM Moved (insertion) [3]

ER 23/10/18 12:54 PM Moved (insertion) [4]

ER 23/10/18 12:54 PM Formatted: Font:Not Italic ER 23/10/18 12:54 PM Formatted: Normal

ER 23/10/18 12:54 PM Deleted: (1976) ER 23/10/18 12:54 PM Deleted: (Hince, 2007) ER 23/10/18 12:54 PM Moved (insertion) [5]

planning software. Digital elevation models (DEMs) of survey sites were compiled using Post Flight Terra 3D 3, and imagery was produced with Global Mapper v16.

To investigate the potential presence of relict glacial geomorphology including moraine features on the inner shelf areas of the New Zealand subantarctic islands, we undertook a ship-based multibeam survey within Perseverance Harbour, Campbell Island (Fig. 1), a valley with an over-deepened U-shaped profile typical of formerly glaciated regions. Multibeam mapping was undertaken with a Kongsberg EM3002 multibeam echo sounder, which was deployed from a side mounted swinging arm. The system is well-suited for detailed seafloor mapping in water depths from less than 1 metre up to typically 200

10

25

metres in cold oceanic conditions. The resultant images were fully georeferenced and simultaneously processed at sub-decimetre accuracy with an integrated differential GPS system.

**2.3 Sedimentary analysis**

Sediment cores were taken from outside the relict glacial limits above Musgrave Harbour, Auckland
 Island (Core 10, Fig. 1, Figs. S1 and S2, Table 1), and from inside ('LLS', Core 14) and outside ('LLS
 Terrace', Core 15) the moraine limits of Lower Lake Speight on Auckland Island (Figs. 1 and 2, Table 1).
 Several other cores were taken from across the Auckland and Cambell Islands (Fig. 1, Table 1) and analysed for basal peat dates. Sediment cores were recovered using a 5-cm wide 'D'-section ('Russian') corer. Organic units at Pillar Rock were sampled with monolith tins. Sediments were described in the

20 field, with samples wrapped and transported back to the laboratory where they were cold-stored (4°C) prior to analysis.

The LLS and LLS Terrance cores were analysed for organic content by loss-on-ignition (LOI). The LLS Cirque core was subsampled (sample size 0.5 cm) every 2 cm from 64-104 cm and every 1 cm thereafter, focussing on the pre-Holocene section of the core. The LLS Terrace core was subsampled (sample size 0.5 cm) every 5 cm from 0-95 cm, and every 2.5 cm thereafter. These samples were dried

| Foi                                                                                                | matted: English (US)                                                                                                                                                                                                                                                                                                                                                                                                                                                                                                                                            |
|----------------------------------------------------------------------------------------------------|-----------------------------------------------------------------------------------------------------------------------------------------------------------------------------------------------------------------------------------------------------------------------------------------------------------------------------------------------------------------------------------------------------------------------------------------------------------------------------------------------------------------------------------------------------------------|
| ER                                                                                                 | 23/10/18 12:54 PM                                                                                                                                                                                                                                                                                                                                                                                                                                                                                                                                               |
| De                                                                                                 | leted:                                                                                                                                                                                                                                                                                                                                                                                                                                                                                                                                                          |
| ER                                                                                                 | 23/10/18 12:54 PM                                                                                                                                                                                                                                                                                                                                                                                                                                                                                                                                               |
| Мо                                                                                                 | ved (insertion) [6]                                                                                                                                                                                                                                                                                                                                                                                                                                                                                                                                             |
| ER                                                                                                 | 23/10/18 12:54 PM                                                                                                                                                                                                                                                                                                                                                                                                                                                                                                                                               |
| Foi                                                                                                | matted: Font:Not Bold, English (AUS)                                                                                                                                                                                                                                                                                                                                                                                                                                                                                                                            |
| ER                                                                                                 | 23/10/18 12:54 PM                                                                                                                                                                                                                                                                                                                                                                                                                                                                                                                                               |
| Foi
Lin                                                                                         | rmatted: Normal, Indent: Left: 0 cm,
e spacing: 1.5 lines                                                                                                                                                                                                                                                                                                                                                                                                                                                                                                    |
| ER                                                                                                 | 23/10/18 12:54 PM                                                                                                                                                                                                                                                                                                                                                                                                                                                                                                                                               |
| Foi                                                                                                | matted: English (AUS)                                                                                                                                                                                                                                                                                                                                                                                                                                                                                                                                           |
| ER                                                                                                 | 23/10/18 12:54 PM                                                                                                                                                                                                                                                                                                                                                                                                                                                                                                                                               |
| For
Out
Sty
Lef
cm                                                                     | matted: Line spacing: 1.5 lines,
line numbered + Level: 2 + Numbering
le: 1, 2, 3, + Start at: 1 + Alignment:
t + Aligned at: 0 cm + Tab after: 0.63
+ Indent at: 0.63 cm                                                                                                                                                                                                                                                                                                                                                                           |
|                                                                                                    |                                                                                                                                                                                                                                                                                                                                                                                                                                                                                                                                                                 |
| / ER                                                                                               | 23/10/18 12:54 PM                                                                                                                                                                                                                                                                                                                                                                                                                                                                                                                                               |
| ER
De                                                                                           | 23/10/18 12:54 PM
leted: a distinct glacial cirque                                                                                                                                                                                                                                                                                                                                                                                                                                                                                                           |
| ER
De
ER                                                                                     | 23/10/18 12:54 PM
leted: a distinct glacial cirque
23/10/18 12:54 PM                                                                                                                                                                                                                                                                                                                                                                                                                                                                               |
| ER
De
ER
De
Spe
cirq
Car
imn
The                                           | 23/10/18 12:54 PM
leted: a distinct glacial cirque
23/10/18 12:54 PM
leted: Named here as Lower 'Lake'
ight (although not itself hosting a lake), the
ue is situated above Coleridge Bay in
nley Harbour, Auckland Island,
nediately south of glacial Lake Speight.
cores are identified as Lower 'Lak([3])                                                                                                                                                                                                                             |
| ER
De
ER
De
Spe
cirq
Car
imn
The
ER                                     | 23/10/18 12:54 PM
leted: a distinct glacial cirque
23/10/18 12:54 PM
leted: Named here as Lower 'Lake'
ight (although not itself hosting a lake), the
ue is situated above Coleridge Bay in
nley Harbour, Auckland Island,
nediately south of glacial Lake Speight.
cores are identified as Lower 'Lake' [3]
23/10/18 12:54 PM                                                                                                                                                                                                       |
| ER
De
Spe
cirq
Car
imn
The
ER
De                                           | 23/10/18 12:54 PM
leted: a distinct glacial cirque
23/10/18 12:54 PM
leted: Named here as Lower 'Lake'
ight (although not itself hosting a lake), the
ue is situated above Coleridge Bay in
nley Harbour, Auckland Island,
nediately south of glacial Lake Speight.
cores are identified as Lower 'Lake' [3]
23/10/18 12:54 PM
leted: In addition, we                                                                                                                                                                             |
| ER
De
ER
De
Spe
cirq
Car
imn
The
ER
De
ER                         | 23/10/18 12:54 PM
leted: a distinct glacial cirque
23/10/18 12:54 PM
leted: . Named here as Lower 'Lake'
ight (although not itself hosting a lake), the
ue is situated above Coleridge Bay in
nley Harbour, Auckland Island,
nediately south of glacial Lake Speight.
cores are identified as Lower 'Lake' [3]
23/10/18 12:54 PM
leted: In addition, we
23/10/18 12:54 PM                                                                                                                                                      |
| ER
De
Spe
cirq
Car
imn
The
ER
De
ER
Mo                               | 23/10/18 12:54 PM
leted: a distinct glacial cirque
23/10/18 12:54 PM
leted: . Named here as Lower 'Lake'
ight (although not itself hosting a lake), the
ue is situated above Coleridge Bay in
nely Harbour, Auckland Island,
nediately south of glacial Lake Speight.
. cores are identified as Lower 'Lake [3]
23/10/18 12:54 PM
leted: In addition, we
23/10/18 12:54 PM
ved up [1]: report here a new ex [4]                                                                                                             |
| ER
De
Spe
cirq
Car
imm
The
ER
De
ER
Mo                               | 23/10/18 12:54 PM
leted: a distinct glacial cirque
23/10/18 12:54 PM
leted: . Named here as Lower 'Lake'
ight (although not itself hosting a lake), the
ue is situated above Coleridge Bay in
nely Harbour, Auckland Island,
nediately south of glacial Lake Speight.
. cores are identified as Lower 'Lake' [3]
23/10/18 12:54 PM
leted: In addition, we
23/10/18 12:54 PM
ved up [1]: report here a new ext [4]
23/10/18 12:54 PM                                                                                      |
| ER
De
Specirq
Carrimm
The
ER
De
ER
Mo
ER
De                          | 23/10/18 12:54 PM
leted: a distinct glacial cirque
23/10/18 12:54 PM
leted: . Named here as Lower 'Lake'
ight (although not itself hosting a lake), the
ue is situated above Coleridge Bay in
nely Harbour, Auckland Island,
nediately south of glacial Lake Speight.
cores are identified as Lower 'Lakc[3]
23/10/18 12:54 PM
leted: In addition, we
23/10/18 12:54 PM
ved up [1]: report here a new exc[4]
23/10/18 12:54 PM
leted: 1 and 3)                                                                        |
| ER
De
Specirq
Carrimn
Thee
ER
Mo
ER
De
ER                               | 23/10/18 12:54 PM
leted: a distinct glacial cirque
23/10/18 12:54 PM
leted: . Named here as Lower 'Lake'
ight (although not itself hosting a lake), the
ue is situated above Coleridge Bay in
nely Harbour, Auckland Island,
nediately south of glacial Lake Speight.
cores are identified as Lower 'Lak([3]
23/10/18 12:54 PM
leted: In addition, we
23/10/18 12:54 PM
ved up [1]: report here a new ex([4])
23/10/18 12:54 PM
leted: 1 and 3)
23/10/18 12:54 PM                                                  |
| ER
De
Specirq
Carrimm
The
ER
De
ER
De
ER
De
ER
De              | 23/10/18 12:54 PM
leted: a distinct glacial cirque
23/10/18 12:54 PM
leted: . Named here as Lower 'Lake'
ight (although not itself hosting a lake), the
ue is situated above Coleridge Bay in
nley Harbour, Auckland Island,
nediately south of glacial Lake Speight.
cores are identified as Lower 'Lak([3]
23/10/18 12:54 PM
leted: In addition, we
23/10/18 12:54 PM
ved up [1]: report here a new ex([4]
23/10/18 12:54 PM
leted: 1 and 3)
23/10/18 12:54 PM
leted: which the organic                       |
| ER
De
Speecirq
Carrimn
The
ER
De
ER
De
ER
De
ER
De
ER       | 23/10/18 12:54 PM
leted: a distinct glacial cirque
23/10/18 12:54 PM
leted: . Named here as Lower 'Lake'
ight (although not itself hosting a lake), the
ue is situated above Coleridge Bay in
nley Harbour, Auckland Island,
nediately south of glacial Lake Speight.
cores are identified as Lower 'Lak([3]
23/10/18 12:54 PM
leted: In addition, we
23/10/18 12:54 PM
ved up [1]: report here a new ex([4])
23/10/18 12:54 PM
leted: 1 and 3)
23/10/18 12:54 PM
leted: which the organic
23/10/18 12:54 PM |
| ER
De
Speciriq
Carrimn
The
ER
De
ER
De
ER
De
ER
De
ER
De | 23/10/18 12:54 PM
leted: a distinct glacial cirque
23/10/18 12:54 PM
leted: . Named here as Lower 'Lake'
ight (although not itself hosting a lake), the
ue is situated above Coleridge Bay in
nley Harbour, Auckland Island,
nediately south of glacial Lake Speight.
cores are identified as Lower 'Lake'[3]
23/10/18 12:54 PM
leted: In addition, we
23/10/18 12:54 PM
leted: 1 and 3)
23/10/18 12:54 PM
leted: which the organic
23/10/18 12:54 PM
leted: Key sedimentary sequences                          |

ER 23/10/18 12:54 PM Deleted: In addition, two

ER 23/10/18 12:54 PM

overnight in an oven at 105°C and then heated in a muffle furnace at 550°C for 4 hours. The LOI was calculated as the percentage dry weight (Heiri et al., 2001).

Two shallow marine sediment cores were taken from Carnley Harbour (50.82°S, 166.01°E, Cores 11 and 12, Fig. 1, Table 1) using a small gravity piston corer with 150cm plastic core barrels deployed over the side of the expedition used. The picton corer has an impact triggered seal that expedies a vacuum.

- the side of the expedition vessel. The piston corer has an impact-triggered seal that creates a vacuum seal on the core barrel once the corer is retrieved, capturing the water sediment interface and preventing 'wash-out' of fine sediment during retrieval. Over 1 m of sediment was recovered from each deployment of the corer, with the cores consisting mostly of peats overlain by fine silts containing occasional marine mollusk shells in life position.
- 10 Samples were taken from the face of the Enderby Formation laminated silt (Fig. 4) at intervals of approximately 10 cm, These were prepared for particle size analysis using a Saturn Digitiser and the results categorised using the Udden-Wentworth Scale (McCave and Syvitski, 1991). The laminated lake sediments from the Enderby Formation were systematically sampled and analysed for pollen, but none was found, in contrast to the earlier study by Fleming et al. (1976)
- 15

20

5

**2.4 Chronology**

2.4.1 Radiocarbon dating

The peat sediments analysed here were given an acid-base-acid (ABA) pretreatment and thencombusted and graphitized in the University of Waikato accelerator mass-spectrometry (AMS) laboratory, with 14C/12C measurement by the University of California at Irvine (UCI) on a NEC compact (1.5SDH) AMS system. The pretreated samples were converted to  $CO_2$  by combustion in sealed prebaked quartz tubes, containing Cu and Ag wire. The  $CO_2$  was then converted to graphite using H2 and a

Fe catalyst, and loaded into aluminium target holders for measurement at UCL. To estimate the timing of the onset of peat growth we used the Phase model option in OxCal v.4.2.4 (Bronk Ramsey and Lee, 2013) with General Outlier analysis detection (probability = 0.05) (Ramsey, 2009). See the extended methodology (Appendix 1) for OxCal code, Importantly, the Phase option is a grouping model which

**ER 23/10/18 12:54 PM**

ER 23/10/18 12:54 PM Deleted: ).

| ER 23/10/18 12:54 PM                                                                                                        | Л                                                                                            |
|-----------------------------------------------------------------------------------------------------------------------------|----------------------------------------------------------------------------------------------|
| Deleted:                                                                                                                    |                                                                                              |
| ER 23/10/18 12:54 PM                                                                                                        | Л                                                                                            |
| Deleted: (Fig. 4).                                                                                                          |                                                                                              |
| ER 23/10/18 12:54 PM                                                                                                        | Л                                                                                            |
| Deleted:                                                                                                                    | [5]                                                                                          |
| ER 23/10/18 12:54 PM                                                                                                        | Л                                                                                            |
| Formatted: Normal                                                                                                           |                                                                                              |
| ER 23/10/18 12:54 PM                                                                                                        | Λ                                                                                            |
| Moved (insertion) [7]                                                                                                       |                                                                                              |
| ER 23/10/18 12:54 PM                                                                                                        | Λ                                                                                            |
| Formatted: English (A                                                                                                       | AUS)                                                                                         |
| ER 23/10/18 12:54 PM                                                                                                        | Λ                                                                                            |
| Formatted: Line space
Outline numbered + L
Style: 1, 2, 3, + Sta
Left + Aligned at: 0 cr
cm + Indent at: 0.63 c | ing: 1.5 lines,
evel: 2 + Numbering
rt at: 1 + Alignment:
n + Tab after: 0.63
cm |

**ER 23/10/18 12:54 PM**

ER 23/10/18 12:54 PM Formatted: Normal ER 23/10/18 12:54 PM Moved (insertion) [8]

assumes no geographic relationship between samples, simply that the ages represent a uniform distribution between a start and end boundary. The 14C ages were calibrated against the Southern Hemisphere calibration (SHCal13) dataset (Hogg et al., 2013) and reported here as mean calendar years (yr) or thousands of years (ka) cal BP  $\pm 1\sigma$  uncertainty (Tables 1 and 2). Radiocarbon ages are distinguished from calibrated ages by being expressed as 14C yr/ka BP.

**2.4.2 Optical dating**

5

10

With no organic material present, to provide an age constraint for the Enderby Formation (Figure 1) we collected four samples (Enderby-1, -2, -3 and -4) for optical dating from the sediments overlying the till, hammering opaque tubes (5 cm in diameter) into the cleaned section faces. The tubes were removed and wrapped in lightproof plastic for transport to the Luminescence Dating Laboratory at the University of Wollongong. Given the potential for the samples to lie beyond the dating range of quartz optically stimulated luminescence, we instead used infrared stimulated luminescence (IRSL), measuring equivalent dose (De) values for individual potassium-rich feldspar (K-feldspar) grains using a pIRIR

15 (post-infrared IRSL) procedure, made possible by recent developments in the field (Buylaert et al., 2009;Li et al., 2014;Li and Li, 2011;Thiel et al., 2011;Thomsen et al., 2008). Details of sample preparation methods are provided in the extended methodology in Appendix 1, together with  $D_e$  and dose rate measurement procedures and supporting data.

**20 2.5 Flowline modelling**

To investigate the impact of changing climatic conditions on glaciation across the southwest Pacific subantarctic islands during the late Pleistocene, we used a glacier flowline model ensemble approach. Due to the complex topography of the Auckland Islands we focussed our modelling on the west-east aligned Perseverance Harbour (Campbell Island), a U-shaped valley likely to have been an independent

25 ice catchment and ideally suited for flowline modelling (Fig. 1D).

| /  | ER 23/10/18 12:54 PM     |
|----|--------------------------|
| /  | Moved (insertion) [9]    |
|    | ER 23/10/18 12:54 PM     |
| /  | Formatted: Font:Italic   |
| ', | ER 23/10/18 12:54 PM     |
| 1  | Formatted: English (AUS) |

ER 23/10/18 12:54 PM Formatted: Line spacing: 1.5 lines, Outline numbered + Level: 3 + Numbering Style: 1, 2, 3, ... + Start at: 1 + Alignment: Left + Aligned at: 0 cm + Tab after: 0.63

**cm + Indent at: 0.63 cm ER 23/10/18 12:54 PM Moved up [2]: ER 23/10/18 12:54 PM Moved up [3]: A new core was therefore taken from Deas Head, and the basal age reported here (Core 4). All samples carbon dated in this study have a carbon conter ... [7] 23/10/18 1 Deleted: (1997) and the upland Mt H .... [8] ER 23/10/18 12:54 PM Moved up [4]: . Here we report new .... [9] ER 23/10/18 12:54 PM Moved down [10]: <#>Optical dating FR 23/10/18 12:54 PI Formatted: Font:Not Bold ER 23/10/18 12:54 PM Deleted: A previously published basa .... [6] Formatted: Font:Not Italic ER 23/10/18 12:54 PM Formatted: English (US) ER 23/10/18 12:54 PM Deleted: ), ER 23/10/18 12:54 PM Deleted: plastic ER 23/10/18 12:54 PM Deleted: (Buylaert et al., 2009; Li et ... [10] ER 23/10/18 12:54 PM Deleted: Supplementary Information FR 23/10/18 12:54 PM Deleted: The optical dating results a ... [11]**

The glacier model employed is essentially the one-dimensional shallow-ice approximation (SIA) flowline model of Golledge & Levy (2011), with two modifications implemented for this study. Firstly, the positive degree-day (PDD) scheme formerly used to calculate surface mass balance (SMB) has been replaced by a simplified energy-balance scheme, following the insolation-temperature melt method (Eq.

5 16 of Robinson et al., (2010)). The second modification to the model is the way in which basal sliding velocities are calculated. Previously we used a triangular averaging scheme (Kamb and Echelmeyer, 1986) to smooth calculated driving stress values, which were then used to calculate sliding velocities using a rate factor and sliding exponent (Eq. 8 of Golledge and Levy (2011)). Here we follow instead the approach of Bueler & Brown (2009), and superpose velocity solutions from the SIA with those from the shallow-shelf approximation (SSA).

10

Input data necessary for our modelling experiments are bed topography (taken from topographic maps) and present-day air temperature and precipitation data (obtained from the National Institute of Water and Atmospheric Research, New Zealand). Time-series data for the study period (125 ka to present) are

15 used for summer and winter insolation values and for air temperature and eustatic sea level

perturbations from present. For air temperature anomalies we use two forcing patterns, both scaled to a range of prescribed minima at the LGM. Given the parallel changes across the Southern Ocean region and the Antarctic (e.g. Röthlisberger et al. (2008); Fogwill et al. (2015)), we use the  $\delta^{18}$ O record from EPICA Dome C (EDC) (Parrenin et al., 2007) as a proxy for Southern Hemisphere atmospheric temperature changes. In addition, we used a proximal sea-surface temperature (SST) record from core

DSDP 594 (Hayward et al., 2008), located to the east of New Zealand's South Island (43.632°S 179.379°E, Fig. 1). Starting at 125 ka we jnitialise our model with present-day topographic and climatological conditions, and simulate glacial advance and retreat according to the calculated SMB and glaciological parameterisation described above.

25

20

Since there are few empirical constraints on glacier extent in Perseverance Harbour, our modelling employs full factorial sampling so that results from the mutual combinations of a range of model and

11

ER 23/10/18 12:54 Deleted: (2011)

ER 23/10/18 12:54 PM Deleted: (2010)). ER 23/10/18 12:54 PM Deleted: (Kamb and Echelmeyer, 1986) ER 23/10/18 12:54 PM

ER 23/10/18 12:54 PM Deleted: NIWA).

ER 23/10/18 12:54 PM Deleted: Fogwill et. al., 2015), ER 23/10/18 12:54 PM Deleted: (Parrenin et al., 2007) ER 23/10/18 12:54 PM

[revised manuscript text omitted]

**ER 23/10/18 12:54 PM Deleted: Middle...iddle of the Pleist ... [12]**

ER 23/10/18 12:54 PM Deleted: extended methodology ER 23/10/18 12:54 PM Formatted: Line spacing: double, Outline numbered + Level: 1 + Numbering Style: 1, 2, 3, ... + Start at: 1 + Alignment: Left + Aligned at: 0 cm + Indent at: 0.63 cm

**ER 23/10/18 12:54 PM**

| Moved up [5]: Using a Sensefly 'eBee'
mapping drone, imagery was acquired, taking
aerial photographs with significant
longitudinal and latitudinal overlaps (80% and
85% respectively) to produce a high-r([14]) |
|-------------------------------------------------------------------------------------------------------------------------------------------------------------------------------------------------------------------------------------|
| ER 23/10/18 12:54 PM                                                                                                                                                                                                                |
| Moved (insertion) [10]                                                                                                                                                                                                              |
| ER 23/10/18 12:54 PM                                                                                                                                                                                                                |
| Deleted: <#>Geomorphological s [13]                                                                                                                                                                                                 |
| ER 23/10/18 12:54 PM                                                                                                                                                                                                                |
| Formatted: English (US)                                                                                                                                                                                                             |
| ER 23/10/18 12:54 PM                                                                                                                                                                                                                |
| Moved up [6]: Multibeam mapping[15]                                                                                                                                                                                                 |
| ER 23/10/18 12:54 PM                                                                                                                                                                                                                |
| Deleted: Lacustrine and peat sedime [17]                                                                                                                                                                                            |
| ER 23/10/18 12:54 PM                                                                                                                                                                                                                |
| Moved up [7]:                                                                                                                                                                                                                       |
| ER 23/10/18 12:54 PM                                                                                                                                                                                                                |
| Deleted: To estimate the timing of tt [18]                                                                                                                                                                                          |
| ER 23/10/18 12:54 PM                                                                                                                                                                                                                |
| Formatted[16]                                                                                                                                                                                                                       |
| ER 23/10/18 12:54 PM                                                                                                                                                                                                                |
| Formatted: English (AUS)                                                                                                                                                                                                            |
| ER 23/10/18 12:54 PM                                                                                                                                                                                                                |
| Moved up [8]: Importantly, the Ph [19]                                                                                                                                                                                              |
| ER 23/10/18 12:54 PM                                                                                                                                                                                                                |
| Formatted[21]                                                                                                                                                                                                                       |
| ER 23/10/18 12:54 PM                                                                                                                                                                                                                |
| Deleted: (Hogg et al., 2013)                                                                                                                                                                                                        |
| ER 23/10/18 12:54 PM                                                                                                                                                                                                                |
| Moved up [9]: and reported here a [20]                                                                                                                                                                                              |
| ER 23/10/18 12:54 PM                                                                                                                                                                                                                |

40 nm, 135 mW/cm2). For the single-grain measurements, we used discs drilled with 100 holes, each 300  $\mu$ m in diameter and depth, and stimulated the grains individually using a focussed infrared laser (830  $\Delta$  10 nm, 400 W/cm2) (Bøtter-Jensen et al., 2003). Radiation doses were given using a 90Sr/90Y beta source mounted on the reader and calibrated for both multi-grain aliquots and individual grain positions.

- 5 The IRSL signals were detected using a photomultiplier tube, after passing through a filter pack containing Schott BG-39 and Corning 7-59 filters to transmit wavelengths of 320–480 nm. We used single aliquots to conduct laboratory tests of anomalous fading, but estimated the *D*e values from measurements of individual grains of K-feldspar (Duller, 2008;Jacobs and Roberts, 2007), enabling the identification and elimination of any grains with aberrant luminescence characteristics (David et al.,
- 10 2007;Feathers, 2003;Feathers et al., 2006;Jacobs et al., 2006;Jacobs et al., 2011;Jacobs et al., 2008;Roberts et al., 1998;Roberts et al., 1999) prior to age determination. We did not apply a residual-dose correction, because our samples have high natural doses (>400 Gy).

In this study, we adopted a two-step single-grain pIRIR regenerative-dose procedure (Blegen et al.,

ER 23/10/18 12:54 PM Deleted: (Bøtter-Jensen et al., 2003)

ER 23/10/18 12:54 PM **Deleted:** (Duller, 2008; Jacobs and Roberts, 2007) ER 23/10/18 12:54 PM

ER 23/10/18 12:54 PM Deleted: (Blegen et al., 2015; Guo et al., 2016)

ER 23/10/18 12:54 PM Deleted: (2012)

ER 23/10/18 12:54 PM **Deleted:** (Thiel et al., 2011)

2015;Guo et al., 2016) to determine the  $D_e$  values for individual sand-sized grains of K-feldspar. In this procedure (see Table S1 for details), an initial infrared stimulation is made at 200 °C for 200 s using infrared diodes, so that all 100 grains on each single-grain disc are stimulated simultaneously. Li and Li (2012) showed that, compared to an initial infrared stimulation at 50 °C, the fading component is removed more effectively at a temperature at 200 °C. The pIRIR signals used for  $D_e$  determination were

15

20 then measured for individual grains at 275 °C, with each grain stimulated by an infrared laser for 1.5 s; this duration proved sufficient to reduce the pIRIR intensity to a low and stable level (Fig. S3). We chose a stimulation temperature of 275 °C, rather than 290 °C (Thiel et al., 2011), to avoid any detrimental effects from significant thermal erosion during the single-grain measurements. The latter can occur at stimulation temperatures close to the preheat temperature (320 °C), because laser stimulation of the final grain on each 100-grain disc begins more than 2.5 min after the first grain on the

5 stimulation of the final grain on each 100-grain disc begins more than 2.5 min after the first grain on the disc. Figure S3 shows a typical pIRIR decay curve and associated dose response curve for a single grain of K-feldspar from Enderby-OSL1.

We conducted a dose recovery test (Galbraith et al., 1999;Roberts et al., 1999) on Enderby-OSL1 to validate the suitability of the pIRIR experimental conditions used to measure these samples (Table S1). A total of 600 grains were bleached for ~6 hr using a Dr Hönle solar simulator (model: UVACUBE

- 400) and then given a dose of ~300 Gy, before being measured using the procedure of Table S1. To select reliable single grains for De determination and reject unsuitable grains, we applied quality-assurance criteria similar to those proposed for single grains of quartz (Jacobs et al., 2006) and K-feldspar (Blegen et al., 2015). Grains were rejected if they exhibited one or more of the following five properties: 1. Weak test-dose signal; 2. High level of recuperation; 3. Poor recycling ratio; 4. Poorly
- fitted dose response curve (DRC); or 5. Natural pIRIR signal equal to or greater than the saturation limit of the DRC. We found that most of the grains (583 grains) were rejected by criterion 1 because of weak  $T_n$  signals. Among the remaining 17 grains, two had  $L_n/T_n$  ratios greater than the saturation limit of the DRC, two had poorly fitted DRCs, and one grain suffered from high recuperation. As a result, only 12 grains were accepted for dose estimation. The measured-to-given dose ratios for these grains are shown
- in Fig. S4a. The weighted mean of these 12 dose recovery ratios is  $1.06 \pm 0.07$ , which is consistent with unity and suggests that the pIRIR procedure in Table S1 yields reliable  $D_e$  values for the Enderby samples. We also tested the Enderby samples for anomalous fading of the pIRIR signal. Twelve aliquots of Enderby-OSL1 were measured using a single-aliquot IRSL procedure similar to that described by Auclair et al. (2003), but based on fading measurements of the pIRIR signal. As our
- 20 samples emitted relatively dim signals, we used the procedure of Thiel et al. (2011), in which the initial infrared stimulation is made at 50 °C and the pIRIR signal is subsequently stimulated at 290 °C. Doses of ~200 Gy were administered using the laboratory beta source, and the irradiated aliquots were then preheated and stored for periods of up to 1 week at room temperature (~20 °C). Figure S4b shows the decay in the sensitivity-corrected pIRIR signal as a function of storage time, normalised to the time of
- 25

ER 23/10/18 12:54 PM Deleted: (Galbraith et al., 1999; Roberts et al., 1999)

ER 23/10/18 12:54 PM Deleted: (Blegen et al., 2015)

ER 23/10/18 12:54 PM Deleted: (2003) ER 23/10/18 12:54 PM Deleted: (2011)

ER 23/10/18 12:54 PM **Deleted:** (Aitken, 1985; Huntley and Lamothe, 2001) ER 23/10/18 12:54 PM **Deleted:** (2012)

prompt measurement. The corresponding fading rate (g-value: (Aitken, 1985;Huntley and Lamothe,

2001) is 0.6  $\pm$  0.7% per decade, which is consistent with zero fading at 1 $\sigma$ . Li and Li (2012)

[revised manuscript text omitted]

23/10/18 12:54 PM Deleted: (Aitken, 1985) ER 23/10/18 12:54 PM Deleted: (Bøtter-Jensen and Mejdahl, 1988; Jacobs and Roberts, 2015) ER 23/10/18 12:54 PM Deleted: (Prescott and Hutton, 1994)

Unknown **Field Code Changed** ER 23/10/18 12:54 Deleted: Huntley and Hancock, 2001; Smedley et al., 2012

ER 23/10/18 12:54 PM Deleted: (2011)

ER 23/10/18 12:54 PM Deleted: (2010)).

ER 23/10/18 12:54 PM Deleted: (2004)

ER 23/10/18 12:54 PM Deleted: (Robinson et al., 2010) prevented from accumulating on slopes steeper than 35° and instead is redistributed (progressively if necessary) to the next downslope cell.

The second modification to the model is the way in which basal sliding velocities are calculated.
Previously we used a triangular averaging scheme (Kamb and Echelmeyer, 1986) to smooth calculated driving stress values, which were then used to calculate sliding velocities using a rate factor and sliding exponent (Eq. 8 of Golledge and Levy (2011)). Here we follow instead the approach of Bueler & Brown (2009), and superpose velocity solutions from the SIA with those from the shallow-shelf approximation (SSA). The latter essentially represents the sliding component of glacier flow, by

10 explicitly incorporating the effects of longitudinal stresses. Our iterative scheme follows MacAyeal (1997): ice thickness and the horizontal velocity gradient are first used to calculate an effective viscosity, from which the horizontal strain rate is determined and then used to update the calculated sliding velocity. We use a coefficient to modify calculated sliding values to account for unknown basal drag.

32

**15**

**3. OxCal code**

Plot() {Curve("ShCal13","ShCal13.14c"); Outlier\_Model("General", T(5), U(0,4), t); 20 Sequence() {Boundary("Start 1"); Phase("1") {R\_Date("Lower Lake Speight Terrace", 12306, 48) {Outlier("General", 0.05); }; 25 R\_Date("O Transect", 12173, 67) {Outlier("General", 0.05);}; R\_Date("Ranui Cove", 9374, 28) {Outlier("General", 0.05);}; 30 R\_Date("Sandy Bay 2", 13020, 36) {Outlier("General", 0.05);}; R\_Date("Deas Head", 11080, 120)

{Outlier("General", 0.05);}; R\_Date("Tagua Bay", 9717, 30) 35 {Outlier("General", 0.05);}; R Date("Red Tape Bog", 11951, 95) {Outlier("General", 0.05);}; R\_Date("Near O on transect", 9057, 52) {Outlier("General", 0.05) }; 40 R\_Date("North Pt Bay", 9292, 98) {Outlier("General", 0.05);}; R Date("Mt Hooker", 10859, 77) {Outlier("General", 0.05);}; R Date("Hooker Cliffs", 12950, 200) 45 {Outlier("General", 0.05);}; R\_Date("Carnley Harbour 3", 10143, 30) {Outlier("General", 0.05);}; R\_Date("Rocky Bay", 11700, 140)

ER 23/10/18 12:54 PM Deleted: (Kamb and Echelmeyer, 1986) ER 23/10/18 12:54 PM Deleted: (2011)).

ER 23/10/18 12:54 PM Deleted: (2009)

ER 23/10/18 12:54 PM Deleted: (1997)

|    | {Outlier("General", 0.05);};                                                                 | 10 | R Date("Homestead Bog Ridge", 12780,          |   |  |  |
|----|-----------------------------------------------------------------------------------------------------|----|-----------------------------------------------|---|--|--|
|    | $\frac{R_{\text{Date}(\text{"Lower Lake Speight", 10150, 40)}}{(Outlier(\text{"Compare1", 0.05)})}$ |    | $\frac{120)}{(Ortlige(  Comparel   = 0.05))}$ |   |  |  |
|    | $\frac{\text{Outher}("General", 0.05); }{\text{R} \text{Date}("Musgrave", 9222, 30)}$               |    |                                               |   |  |  |
| 5  | {Outlier("General", 0.05):}:                                                                        |    | {Outlier("General", 0.05);}:                  |   |  |  |
|    | R_Date("Pillar Rock", 10086, 31)                                                                    | 15 | R Date("Mt Honey Saddle", 12445, 76)          |   |  |  |
|    | {Outlier("General", 0.05);};                                                                        |    | {Outlier("General", 0.05);};                  |   |  |  |
|    | R_Date("Carnley core 2", 10681, 29)                                                          |    | R_Date("Col Ridge", 9416, 57)                 |   |  |  |
|    | {Outlier("General", 0.05);};                                                                        |    | {Outlier("General", 0.05);};};                |   |  |  |
|    |                                                                                                     |    | Boundary("End 1");};};                        | • |  |  |
| 20 |                                                                                                     |    |                                               |   |  |  |

ER 23/10/18 12:54 PM

**Data availability**

Data will be lodged on the NOAA Paleoclimate Archive

**Acknowledgements**

[revised manuscript text omitted]

ER 23/10/18 12:54 PM

ER 23/10/18 12:54 PM

| 1 | ER 23/10/18 12:54 PM         |
|---|------------------------------|
|   | Deleted: .                   |
|   | ER 23/10/18 12:54 PM         |
|   | Deleted: -383, 2009a. |
|   | ER 23/10/18 12:54 PM         |
|   | Deleted: .                   |
|   |                              |

| Moved (insertion) [14]                                                                                                                                                            |
|-----------------------------------------------------------------------------------------------------------------------------------------------------------------------------------|
| ER 23/10/18 12:54 PM                                                                                                                                                              |
| Moved (insertion) [15]                                                                                                                                                            |
| ER 23/10/18 12:54 PM                                                                                                                                                              |
| Moved up [14]: and Mejdahl, V.:
Assessment of beta dose-rate using a GM
multicounter system, International Journal of
Radiation Applications and Instrumentation. |
| ER 23/10/18 12:54 PM                                                                                                                                                              |
| Moved up [15]: Part D. Nuclear Tracks
and Radiation Measurements, 14, 187-191,
1988.                                                                                        |
| ER 23/10/18 12:54 PM                                                                                                                                                              |
| Deleted: Bøtter-Jensen, L.                                                                                                                                                        |
| ER 23/10/18 12:54 PM                                                                                                                                                              |
| Deleted: .                                                                                                                                                                        |
| ER 23/10/18 12:54 PM                                                                                                                                                              |

[revised manuscript text omitted]

- Quaternary Science Reviews, 27, 284-294, 10.1016/j.quascirev.2007.09.013, 2008. Kohfeld, K., Graham, R., De Boer, A., Sime, L., Wolff, E., Le Quéré, C., and Bopp, L.: Southern Hemisphere westerly wind changes during the Last Glacial Maximum: paleo-data synthesis, Quaternary Science Reviews, 68, 76-95, 2013.
- Laskar, J., Robutel, P., Joutel, F., Gastineau, M., Correia, A., and Levrard, B.: A long-term numerical solution for the insolation quantities of the Earth, Astronomy & Astrophysics, 428, 261-285, 2004.
   Li, B., and Li, S.-H.: Luminescence dating of K-feldspar from sediments: a protocol without anomalous fading correction, Quaternary Geochronology, 6, 468-479, 2011.
- 20 Li, B., and Li, S.: A reply to the comments by Thomsen et al. on" Luminescence dating of K-feldspar from sediments: a protocol without anomalous fading correction", 2012.
- Li, B., Jacobs, Z., Roberts, R. G., and Li, S.-H.: Review and assessment of the potential of post-IR IRSL dating methods to circumvent the problem of anomalous fading in feldspar luminescence, Geochronometria, 41, 178-201, 2014.
- Li, B., Jacobs, Z., and Roberts, R. G.: Investigation of the applicability of standardised growth curves
   for OSL dating of quartz from Haua Fteah cave, Libya, Quaternary Geochronology, 35, 1-15, 2016.
   Li, B., Jacobs, Z., Roberts, R. G., Galbraith, R., and Peng, J.: Variability in quartz OSL signals caused
   by measurement uncertainties: Problems and solutions, Quaternary Geochronology, 41, 11-25, 2017.
- Li, W., Wang, R., Xiang, F., Ding, X., and Zhao, M.: Sea surface temperature and subtropical front movement in the South Tasman Sea during the last 800 ka, Chinese Science Bulletin, 55, 3338-3344, 2010.
- Lowe, J4.2 and Walker, M.: Radiocarbon dating the last glacial-interglacial transition (Ca. 14-9 14C ka BP) in terrestrial and marine records: the need for new quality assurance protocols, Radiocarbon, 42, 53-68, 2000.
- MacAyeal, D. R.: EISMINT: Lessons in ice-sheet modeling, Department of Geophysical Sciences, J5 University of Chicago, Chicago, IL, 1832, 1839, 1997.
- Marshall, J4, and Speer, K.: Closure of the meridional overturning circulation through Southern Ocean upwelling, Nature Geoscience, 5, 171-180, 10.1038/ngeo1391, 2012.
- McGlone, M., Wilmshurst, J., and Meurk, C.: Climate, fire, farming and the recent vegetation history of subantarctic Campbell Island, Earth and Environmental Science Transactions of the Royal Society of
   Edinburgh, 98, 71-84, 2007.

ER 23/10/18 12:54 PN

**Moved up [18]:** ., Wintle, A. G., Duller, G. A., Roberts, R. G., and Wadley, L.: New ages for the post-Howiesons Poort, late and final Middle stone age at Sibudu, South Africa, Journal of Archaeological Science, 35, 1790-1807, 2008.

| ER 23/10/18 12:54 PM                                                                                                                                                                                                                                                                                                                                                                                                                                                                                                                                                                                                                                                                                            |
|-----------------------------------------------------------------------------------------------------------------------------------------------------------------------------------------------------------------------------------------------------------------------------------------------------------------------------------------------------------------------------------------------------------------------------------------------------------------------------------------------------------------------------------------------------------------------------------------------------------------------------------------------------------------------------------------------------------------|
| Deleted: .                                                                                                                                                                                                                                                                                                                                                                                                                                                                                                                                                                                                                                                                                                      |
| ER 23/10/18 12:54 PM                                                                                                                                                                                                                                                                                                                                                                                                                                                                                                                                                                                                                                                                                            |
| Moved (insertion) [19]                                                                                                                                                                                                                                                                                                                                                                                                                                                                                                                                                                                                                                                                                          |
| ER 23/10/18 12:54 PM                                                                                                                                                                                                                                                                                                                                                                                                                                                                                                                                                                                                                                                                                            |
| Moved (insertion) [20]                                                                                                                                                                                                                                                                                                                                                                                                                                                                                                                                                                                                                                                                                          |
| ER 23/10/18 12:54 PM                                                                                                                                                                                                                                                                                                                                                                                                                                                                                                                                                                                                                                                                                            |
| Moved (insertion) [21]                                                                                                                                                                                                                                                                                                                                                                                                                                                                                                                                                                                                                                                                                          |
| ER 23/10/18 12:54 PM                                                                                                                                                                                                                                                                                                                                                                                                                                                                                                                                                                                                                                                                                            |
| Moved (insertion) [22]                                                                                                                                                                                                                                                                                                                                                                                                                                                                                                                                                                                                                                                                                          |
| ER 23/10/18 12:54 PM                                                                                                                                                                                                                                                                                                                                                                                                                                                                                                                                                                                                                                                                                            |
| Moved up [22]: B., Jacobs, Z., Roberts, R.
G., and Li, SH.: Review and assessment of
the potential of post-IR IRSL dating methods
to circumvent the problem of anomalous
fading in feldspar luminescence,
Geochronometria, 41, 178-201, 2014.                                                                                                                                                                                                                                                                                                                                                                                                                                             |
| ER 23/10/18 12:54 PM                                                                                                                                                                                                                                                                                                                                                                                                                                                                                                                                                                                                                                                                                            |
| Deleted:                                                                                                                                                                                                                                                                                                                                                                                                                                                                                                                                                                                                                                                                                                        |
| Deleteu.                                                                                                                                                                                                                                                                                                                                                                                                                                                                                                                                                                                                                                                                                                        |
| ER 23/10/18 12:54 PM                                                                                                                                                                                                                                                                                                                                                                                                                                                                                                                                                                                                                                                                                            |
| ER 23/10/18 12:54 PM
Moved up [20]: and Li, S.: A reply to the
comments by Thomsen et al. on" Lumi                                                                                                                                                                                                                                                                                                                                                                                                                                                                                                                                                                                                        |
| ER 23/10/18 12:54 PM
Moved up [20]: and Li, S.: A reply to the
comments by Thomsen et al. on" Lumi
ER 23/10/18 12:54 PM                                                                                                                                                                                                                                                                                                                                                                                                                                                                                                                                                                                |
| ER 23/10/18 12:54 PM
Moved up [20]: and Li, S.: A reply to the
comments by Thomsen et al. on" Lumi
ER 23/10/18 12:54 PM
Moved up [21]: on" Luminescence dating
of K-feldspar from sediments: a protocol
without anomalous fading correction", 2012.                                                                                                                                                                                                                                                                                                                                                                                                                                           |
| ER 23/10/18 12:54 PM
Moved up [20]: and Li, S.: A reply to the
comments by Thomsen et al. on" Lumi
ER 23/10/18 12:54 PM
Moved up [21]: on" Luminescence dating
of K-feldspar from sediments: a protocol
without anomalous fading correction", 2012.
ER 23/10/18 12:54 PM                                                                                                                                                                                                                                                                                                                                                                                                                   |
| ER 23/10/18 12:54 PM         Moved up [20]: and Li, S.: A reply to the comments by Thomsen et al. on" Lumi         ER 23/10/18 12:54 PM         Moved up [21]: on" Luminescence dating of K-feldspar from sediments: a protocol without anomalous fading correction", 2012.         ER 23/10/18 12:54 PM         Deleted: 2012                                                                                                                                                                                                                                                                                                                                                                                  |
| ER 23/10/18 12:54 PM
Moved up [20]: and Li, S.: A reply to the
comments by Thomsen et al. on" Lumi
ER 23/10/18 12:54 PM
Moved up [21]: on" Luminescence dating
of K-feldspar from sediments: a protocol
without anomalous fading correction", 2012.
ER 23/10/18 12:54 PM
Deleted: 2012                                                                                                                                                                                                                                                                                                                                                                                                  |
| ER 23/10/18 12:54 PM         Moved up [20]: and Li, S.: A reply to the comments by Thomsen et al. on" Lumi         ER 23/10/18 12:54 PM         Moved up [21]: on" Luminescence dating of K-feldspar from sediments: a protocol without anomalous fading correction", 2012.         ER 23/10/18 12:54 PM         Deleted: 2012.      [22]         ER 23/10/18 12:54 PM         Deleted: 2012.      [22]         ER 23/10/18 12:54 PM         Deleted: 2012.      [22]         ER 23/10/18 12:54 PM         Moved up [19]: and Li, SH.:         Luminescence dating of K-feldspar from sediments: a protocol without anomale([23])                                                                               |
| ER 23/10/18 12:54 PM
Moved up [20]: and Li, S.: A reply to the
comments by Thomsen et al. on" Lumi
ER 23/10/18 12:54 PM
Moved up [21]: on" Luminescence dating
of K-feldspar from sediments: a protocol
without anomalous fading correction", 2012.
ER 23/10/18 12:54 PM
ER 23/10/18 12:54 PM
Moved up [19]: and Li, SH.:
Luminescence dating of K-feldspar from
sediments: a protocol without anomal«[23]
ER 23/10/18 12:54 PM                                                                                                                                                                                                                        |
| ER 23/10/18 12:54 PM         Moved up [20]: and Li, S.: A reply to the comments by Thomsen et al. on" Lumi         ER 23/10/18 12:54 PM         Moved up [21]: on" Luminescence dating of K-feldspar from sediments: a protocol without anomalous fading correction", 2012.         ER 23/10/18 12:54 PM         Deleted: 2012.      [22]         ER 23/10/18 12:54 PM         Deleted: 2012.      [22]         ER 23/10/18 12:54 PM         Moved up [19]: and Li, SH.:         Luminescence dating of K-feldspar from sediments: a protocol without anomal         sediments: a protocol without anomal         ER 23/10/18 12:54 PM         Deleted: 2012.         ER 23/10/18 12:54 PM         Deleted: Li, |
| ER 23/10/18 12:54 PM

[revised manuscript text omitted]

- Winnshutst, J. H., Fenwick, F., Zuliz, Y., Goosse, H., Wilson, K. J., Calet, E., Elpson, W., Jones, R. T., Harsch, M., Clark, G., Marzinelli, E., Rogers, T., Rainsley, E., Ciasto, L., Waterman, S., Thomas, E. R., and Visbeck, M.: Tropical forcing of increased Southern Ocean climate variability revealed by a 140-year subantarctic temperature reconstruction, Clim. Past, 13, 231-248, 10.5194/cp-13-231-2017,
- 2017. Van der Putten, N., Verbruggen, C., Ochyra, R., Verleyen, E., and Frenot, Y.: Subantarctic flowering plants: pre-glacial survivors or post-glacial immigrants?, Journal of Biogeography, 37, 582-592, 2010.
- WAIS, D. P. M.: Precise interpolar phasing of abrupt climate change during the last ice age, Nature,
   520, 661-665, 10.1038/nature14401
  - http://www.nature.com/nature/journal/v520/n7549/abs/nature14401.html#supplementary-information, 2015.
- Walker, M., Johnsen, S., Rasmussen, S. O., Popp, T., Steffensen, J.-P., Gibbard, P., Hoek, W., Lowe, J.,
  Andrews, J., Bjorck, S., Cwynar, L. C., Hughen, K., Kershaw, P., Kromer, B., Litt, T., Lowe, D. J.,

41

ER 23/10/18 12:54 PM Moved (insertion) [25]

ER 23/10/18 12:54 PM Deleted: Thomsen, K. J., Murray, A. S., ER 23/10/18 12:54 PM

**Moved up [25]:** Jain, M., and Bøtter-Jensen, L.: Laboratory fading rates of various luminescence signals from feldspar-rich sediment extracts, Radiation measurements, 43, 1474-1486, 2008.

ER 23/10/18 12:54 PM Deleted: Turney, C. S

**ER 23/10/18 12:54 PM**

**Moved down [26]:** ., Jones, R. T., Lister, D., Jones, P., Williams, A. N., Hogg, A., Thomas, Z. A., Compo, G. P., Yin, X., and Fogwill, C. J.: Anomalous mid-twentieth century atmospheric circulation change over the South Atlantic compared to the last 6000 years,

ER 23/10/18 12:54 PM

ER 23/10/18 12:54 PM

Moved (insertion) [26] ER 23/10/18 12:54 PM

ER 23/10/18 12:54 PM

ER 23/10/18 12:54 PM

Nakagawa, T., Newnham, R., and Schwander, J.: Formal definition and dating of the GSSP (Global Stratotype Section and Point) for the base of the Holocene using the Greenland NGRIP ice core, and selected auxiliary records, Journal of Quaternary Science, 24, 3-17, 10.1002/jqs.1227, 2009.

Watson, E.: Two nivation cirques near Aberystwyth, Wales, Biuletyn Peryglacjalny, 15, 79-101, 1966.
Williams, P. W., McGlone, M., Neil, H., and Zhao, J.-X.: A review of New Zealand palaeoclimate from the Last Interglacial to the global Last Glacial Maximum, Quaternary Science Reviews, 110, 92-106, http://dx.doi.org/10.1016/j.quascirev.2014.12.017, 2015.

ER 23/10/18 12:54 PM Deleted: 2015.

|    |          |                              |                |                 |         |                                        |                |               |                  |        |              |     |              | /                                            | ER 23/10/18 12:54 PM |
|----|----------|------------------------------|----------------|-----------------|---------|----------------------------------------|----------------|---------------|------------------|--------|--------------|-----|--------------|----------------------------------------------|----------------------|
|    | ч        | Fables                       |                |                 |         |                                        |                |               |                  |        |              |     |              |                                              | Deleted: 15471       |
|    |          | lables                       |                |                 |         |                                        |                |               |                  |        |              |     |              |                                              | ER 23/10/18 12:54 PM |
|    |          |                              |                |                 |         | LTEL                                   |                |               |                  |        | Uncalibrated |     | Calibrated   |                                              | Deleted: 107         |
| #  | Location | Site                         | Lat (S)        | Long (E)        | Elevati | Lab                                    | Lab            | Depth
(cm) | Material         | 4120   | age (14C yrs | ±/- | mean age     |                                              | ER 23/10/18 12:54 PM |
| -  | Location | Site                         | Lat (5)        | Long (L)        | on (m)  | coue                                   | Wk-            | (ciii)        | wateria          | uise   | Dr)          | +/- | (913 DF)     |                                              | Deleted: 10439       |
| 1  | Auckand  | Sandy Bay 2                  | 50°29'55.90"S  | 166°17'23.80"E  | 5       | X13/82                                 | 38429          | 119           | Peat             | n.m    | 13020        | 36  | 15502        | 206                                          | ER 23/10/18 12:54 PM |
| 2  | Auckand  | North Point Bay              | 50°31'0.09"S   | 166°10'29.60"E  | 40      | X94/16                                 | NZ
8226     | 45            | Wood             | -28.8  | 9292         | 98  | 10440        | 131                                          | Deleted: 125         |
| 2  |          | Pillar Rock                  | 50820150 20110 | 400842150.07115 | 20      | V4 4/52                                | WK-            | 440           | Deet             |        | 10000        | 24  | 11510        |                                              | ER 23/10/18 12:54 PM |
| 3  | Auckand  | Exposure                     | 50-30 58.29 5  | 100°12'59.07"E  | 20

---

## Author Response (AR2)

**Response to referees**

We thank the referees for their further comments, and for their acknowledgement of the positive changes we have made to this manuscript. We respond to their comments below.

**Referee 1**

**-Why do the authors used the SST reconstruction by Hayward et al. for ODP 594 and not the higher resolution record from the same site: core MD97-2120 by Pahnke et al.? Furthermore in fig. 9 they plot the Pahnke et al. temperature record instead of the Hayward et al. record they used for part of the flowline model runs. In the same figure they could also show the Hayward temperature record over the last 500ka indicating local temperature changes for stage 10 and 12.**

We acknowledge that the choice of proxy used here could potentially produce different glacier behaviour in our model, and have added a statement to this effect to the methodology. However, both the Hayward and Pahnke records closely follow large-scale changes both in each other and in the EPICA Dome C record, suggesting that any differences would be minor. It is important to note that the model run that fit most closely our geomorphological constraints (Figure 8A) was forced using the EPICA Dome C temperature record, not the SST record, and so choice of proxy reconstruction here does not affect our conclusions either way.

**I would appreciate that the authors indicate why they choose to run the model over the last glacial cycle and not the last 4 cycles: computing time? Problems to reconstitute the bed topography back in time?**

The referee is correct in stating that computational costs made modelling further back in time impractical. Furthermore, an ensemble such as this run over a longer period would have led to increasing uncertainties associated with all physical boundary conditions. We have clarified this in the methodology.

**The data indicate a glaciation with an larger extent than the LGM one that the authors assume to occur around 68ka. They do not discuss the fact that it could be the maximum extent during stage 6 or stage 8. I think a few lines at least are necessary to rule out these possibilities.**
Thanks to Referee 1 for this suggestion – we have added some discussion around this to Section 4.4.

**A number of typo errors are to be corrected, a few of them are listed below.**
**2.1 Line 26 : Peninsula n missing**
**3.2 line 25 suppress "is"**
**4.1 line 15 "separate" + correct end of line**
**p20 Line 21 "formation"**

Corrected, thank you.

**Fig 7 What are the two yellow at ~55 et 63kans?**

These yellow 'bars' are in fact an artefact of their being several of the 288 model runs that overlap at these points. We have altered the weight of these lines to try and alleviate the impression of a bar.

**Fig. 9 "Basal peat age range" interval is not placed correctly on the figure.**

Corrected, thank you.

**The ACR is not marked as a strong SST decrease in the SST records they present nor in other SST records of the Southern Ocean from the Subantarctic zone, opposite to what the authors says. SST temperature records are different than EDC isotopic record, they indicate temperature closer to Holocene ones before the ACR and indicate a small decrease or on only a plateau in the warming for the ACR.**

We would respectively disagree with the first statement here – Figure 9 clearly shows a marked decrease in SSTs during the ACR during an otherwise upwards trend following the LGM. The SST and EDC records to indeed differ in parts, although broad-scale trends are the same. This Is a reflection of the fact that the EDC record reflects atmospheric rather than ocean temperatures, which respond in different ways to external forcings.

**Referee 2**

**I would suggest the following improvements:**
**- The age range of the samples Enderby-1, -2, and -4 are shown now on different levels. This does not make sense, as they are not representing different SST. I would suggest adding the ages on top of the SST graph. Maybe add in bold also the weighted-mean age 378 +/- 26 ka. In this way it gets clear that the authors attribute the Enderby Glaciation to MIS10.**
**- Even it is an interpretation of the authors I would add the Perseverance Harbour moraine, which was formed around 68 ka (based on the modeled glacier maximum extent) and the formation of the circle moraines interpreted to be of eLGM/LGM age. Maybe write these two features in italics as they are not directly dated.**
**This would make Figure 9 even stronger and would serve as a compilation of the various climate/glacier events.**

Many thanks to the referee for their further suggestion to improve Figure 9; where practical we have made these changes.

**Small technical comments:**
**p. 15, line 6: The age error in the text was changed to +- 120 years, but in Fig. 3 it is still +-110**
**p. 18, line 19: add "Argentina" to the location Lago Buenos Aires**

Changed, thank you.

[revised manuscript text omitted]

R_Date("Mt Hooker", 10859, 77)                         R_Date("Carnley core 2", 10681, 29)
{Outlier("General", 0.05);};                            {Outlier("General", 0.05);};
R_Date("Hooker Cliffs", 12950, 200)                     R_Date("Homestead Bog Ridge", 12780,
5    {Outlier("General", 0.05);};                        120)
R_Date("Carnley Harbour 3", 10143, 30)          20    {Outlier("General", 0.05);};
{Outlier("General", 0.05);};                            R_Date("Homestead Scarp", 13648, 73)
R_Date("Rocky Bay", 11700, 140)                         {Outlier("General", 0.05);};
{Outlier("General", 0.05);};                            R_Date("Mt Honey Saddle", 12445, 76)
10   R_Date("Lower Lake Speight", 10150, 40)            {Outlier("General", 0.05);};
{Outlier("General", 0.05);};                     25    R_Date("Col Ridge", 9416, 57)
R_Date("Musgrave", 9222, 30)                            {Outlier("General", 0.05);};};};
{Outlier("General", 0.05);};                            Boundary("End 1");};};
R_Date("Pillar Rock", 10086, 31)
```

[revised manuscript text omitted]

Eleanor Fogwill 14/12/18 8:57 AM

Eleanor Fogwill 14/12/18 8:57 AM

---

## Author Response (AR3)

Many thanks to the reviewers and the editor for their input and improvements. Please find below a marked-up copy in which we highlight the change in case to the Porter reference.

[revised manuscript text omitted]

Eleanor Fogwill 19/12/18 12:50 PM
**Comment [1]:** Porter reference changed from upper case